# IGLU: Efficient GCN Training via Lazy Updates

**S Deepak Narayanan**[*][†] **& Aditya Sinha**[*][‡]
Microsoft Research India
{sdeepaknarayanan1,adityaasinha28}@gmail.com

**Prateek Jain**[‡]
Microsoft Research India
prajain@google.com

**Purushottam Kar**
IIT Kanpur & Microsoft Research India
purushot@cse.iitk.ac.in

**Sundararajan Sellamanickam**
Microsoft Research India
ssrajan@microsoft.com

## Abstract

Training multi-layer Graph Convolution Networks (GCN) using standard SGD techniques scales poorly as each descent step ends up updating node embeddings for a large portion of the graph. Recent attempts to remedy this sub-sample the graph that reduce compute but introduce additional variance and may offer suboptimal performance. This paper develops the IGLU method that caches intermediate computations at various GCN layers thus enabling lazy updates that significantly reduce the compute cost of descent. IGLU introduces bounded bias into the gradients but nevertheless converges to a first-order saddle point under standard assumptions such as objective smoothness. Benchmark experiments show that IGLU offers up to 1.2% better accuracy despite requiring up to 88% less compute.

## 1 Introduction

The Graph Convolution Network (GCN) model is an effective graph representation learning technique. Its ability to exploit network topology offers superior performance in several applications such as node classification (Kipf & Welling, 2017), recommendation systems (Ying et al., 2020) and program repair (Yasunaga & Liang, 2020). However, training multi-layer GCNs on large and dense graphs remains challenging due to the very aggregation operation that enables GCNs to adapt to graph topology – a node's output layer embedding depends on embeddings of its neighbors in the previous layer which recursively depend on embeddings of their neighbors in the previous layer, and so on. Even in GCNs with 2-3 layers, this prompts back propagation on loss terms for a small mini-batch of nodes to update a large multi-hop neighborhood causing mini-batch SGD techniques to scale poorly.

Efforts to overcome this problem try to limit the number of nodes that receive updates as a result of a back-propagation step Chiang et al. (2019); Hamilton et al. (2017); Zeng et al. (2020). This is done either by sub-sampling the neighborhood or clustering (it is important to note the distinction between nodes sampled to create a mini-batch and neighborhood sampling done to limit the neighborhood of the mini-batch that receives updates). Variance reduction techniques Chen et al. (2018a) attempt to reduce the additional variance introduced by neighborhood sampling. However, these techniques often require heavy subsampling in large graphs resulting in poor accuracy due to insufficient aggregation. They also do not guarantee unbiased learning or rigorous convergence guarantees. See Section 2 for a more detailed discussion on the state-of-the-art in GCN training.

**Our Contributions**: This paper presents IGLU, an efficient technique for training GCNs based on lazy updates. An analysis of the gradient structure in GCNs reveals the most expensive component of the back-propagation step initiated at a node to be (re-)computation of forward-pass embeddings for its vast multi-hop neighborhood. Based on this observation, IGLU performs back-propagation with significantly reduced complexity using intermediate computations that are cached at regular intervals.

---

[*]Authors contributed equally
[†]Now at ETH Zurich
[‡]Now at Google Research India

This completely avoids neighborhood sampling and is a stark departure from the state-of-the-art. IGLU is architecture-agnostic and can be readily implemented on a wide range of GCN architectures. Avoiding neighborhood sampling also allows IGLU to completely avoid variance artifacts and offer provable convergence to a first-order stationary point under standard assumptions. In experiments, IGLU offered superior accuracies and accelerated convergence on a range of benchmark datasets.

## 2 RELATED WORKS

(Bruna et al., 2014; Defferrard et al., 2016; Kipf & Welling, 2017) introduced the GCN architecture for transductive learning on graphs. Later works extended to inductive settings and explored architectural variants such as the GIN (Xu et al., 2019). Much effort has focused on speeding-up GCN training.

**Sampling Based Approaches**: The *neighborhood sampling* strategy e.g. GraphSAGE (Hamilton et al., 2017) limits compute by restricting back-propagation updates to a sub-sampled neighborhood of a node. *Layer sampling* strategies such as FastGCN (Chen et al., 2018b), LADIES (Zou et al., 2019) and ASGCN (Huang et al., 2018) instead sample nodes at each GCN layer using importance sampling to reduce variance and improve connectivity among sampled nodes. FastGCN uses the same sampling distribution for all layers and struggles to maintain connectivity unless large batch-sizes are used. LADIES uses a per-layer distribution conditioned on nodes sampled for the succeeding layer. ASGCN uses a linear model to jointly infer node importance weights. Recent works such as Cluster-GCN (Chiang et al., 2019) and GraphSAINT (Zeng et al., 2020) propose *subgraph sampling* creating mini-batches out of subgraphs and restricting back-propagation to nodes within the subgraph. To avoid losing too many edges, large mini-batch sizes are used. Cluster-GCN performs graph clustering and chooses multiple clusters per mini-batch (reinserting any edges cutting across clusters in a mini-batch) whereas GraphSAINT samples large subgraphs directly using random walks.

**Bias and Variance**: Sampling techniques introduce bias as non-linear activations in the GCN architecture make it difficult to offer unbiased estimates of the loss function. Zeng et al. (2020) offer unbiased estimates if non-linearities are discarded. Sampling techniques also face increased variance for which variance-reduction techniques have been proposed such as VR-GCN (Chen et al., 2018a), MVS-GNN (Cong et al., 2020) and AS-GCN (Huang et al., 2018). VR-GCN samples nodes at each layer whose embeddings are updated and uses stale embeddings for the rest, offering variance elimination in the limit under suitable conditions. MVS-GNN handles variance due to mini-batch creation by performing importance weighted sampling to construct mini-batches. The Bandit Sampler (Liu et al., 2020) formulates variance reduction as an adversarial bandit problem.

**Other Approaches:** Recent approaches decouple propagation from prediction as a pre-processing step e.g. PPRGo (Bojchevski et al., 2020), APPNP (Klicpera et al., 2019) and SIGN (Frasca et al., 2020). APPNP makes use of the relationship between the GCNs and PageRank to construct improved propagation schemes via personalized PageRank. PPRGo extends APPNP by approximating the dense propagation matrix via the push-flow algorithm. SIGN proposes inception style pre-computation of graph convolutional filters to speed up training and inference. GNNAutoScale (Fey et al., 2021) builds on VR-GCN and makes use of historical embeddings for scaling GNN training to large graphs.

**IGLU in Context of Related Work**: IGLU avoids neighborhood sampling entirely and instead speeds-up learning using stale computations. Intermediate computations are cached and lazily updated at regular intervals e.g. once per epoch. We note that IGLU's *caching* is distinct and much more aggressive (e.g. lasting an entire epoch) than the internal caching performed by popular frameworks such as TensorFlow and PyTorch (where caches last only a single iteration). Refreshing these caches in bulk offers IGLU economies of scale. IGLU incurs no sampling variance but incurs bias due to the use of stale computations. Fortunately, this bias is provably bounded, and can be made arbitrarily small by adjusting the step length and refresh frequency of the stale computations.

## 3 IGLU: EFFICIENT GCN TRAINING VIA LAZY UPDATES

**Problem Statement**: Consider the problem of learning a GCN architecture on an undirected graph $\mathcal{G}(\mathcal{V}, \mathcal{E})$ with each of the $N$ nodes endowed with an *initial* feature vector $\mathbf{x}_i^0 \in \mathbb{R}^{d_0}, i \in \mathcal{V}$. $X^0 \in \mathbb{R}^{n \times d_0}$ denotes the matrix of these initial features stacked together. $\mathcal{N}(i) \subset \mathcal{V}$ denotes the set of neighbors of node $i$. $A$ denotes the (normalized) adjacency matrix of the graph. A multi-layer GCN

architecture uses a parameterized function at each layer to construct a node's embedding for the next layer using embeddings of that node as well as those of its neighbors. Specifically

$$\mathbf{x}_i^k = f(\mathbf{x}_j^{k-1}, j \in \{i\} \cup \mathcal{N}(i); E^k),$$

where $E^k$ denotes the parameters of $k$-th layer. For example, a classical GCN layer is given by

$$\mathbf{x}_i^k = \sigma \left( \sum_{j \in \mathcal{V}} A_{ij}(W^k)^\top \mathbf{x}_j^{k-1} \right),$$

where $E^k$ is simply the matrix $W^k \in \mathbb{R}^{d_{k-1} \times d_k}$ and $d_k$ is the embedding dimensionality at the $k^{\text{th}}$ layer. IGLU supports more involved architectures including residual connections, virtual nodes, layer normalization, batch normalization, etc (see Appendix A.5). We will use $E^k$ to collectively refer to all parameters of the $k^{\text{th}}$ layer e.g. offset and scale parameters in a layer norm operation, etc. $X^k \in \mathbb{R}^{n \times d_k}$ will denote the matrix of $k^{\text{th}}$ layer embeddings stacked together, giving us the handy shorthand $X^k = f(X^{k-1}; E^k)$. Given a $K$-layer GCN and a multi-label/multi-class task with $C$ labels/classes, a fully-connected layer $W^{K+1} \in \mathbb{R}^{d_K \times C}$ and activation functions such as sigmoid or softmax are used to get predictions that are fed into the task loss. IGLU does not require the task loss to decompose over the classes. The convergence proofs only require a smooth training objective.

**Neighborhood Explosion**: To understand the reasons behind neighborhood explosion and the high cost of mini-batch based SGD training, consider a toy univariate regression problem with unidimensional features and a 2-layer GCN with sigmoidal activation i.e. $K = 2$ and $C = 1 = d_0 = d_1 = d_2$. This GCN is parameterized by $w^1, w^2, w^3 \in \mathbb{R}$ and offers the output $\hat{y}_i = w^3 \sigma\left(z_i^2\right)$ where $z_i^2 = \sum_{j \in \mathcal{V}} A_{ij} w^2 x_j^1 \in \mathbb{R}$. In turn, we have $x_j^1 = \sigma\left(z_i^1\right)$ where $z_i^1 = \sum_{j' \in \mathcal{V}} A_{jj'} w^1 x_{j'}^0, \in \mathbb{R}$ and $x_{j'}^0, \in \mathbb{R}$ are the initial features of the nodes. Given a task loss $\ell : \mathbb{R} \times \mathbb{R} \to \mathbb{R}_+$ e.g. least squares, denoting $\ell_i' = \ell'(\hat{y}_i, y_i)$ gives us

$$\frac{\partial \ell(\hat{y}_i, y_i)}{\partial w^1} = \ell_i' \cdot \frac{\partial \hat{y}_i}{\partial z_i^2} \cdot \frac{\partial z_i^2}{\partial w^1} = \ell_i' \cdot w^3 \sigma'(z_i^2) \cdot \sum_{j \in \mathcal{V}} A_{ij} w^2 \frac{\partial x_j^1}{\partial w^1}$$

$$= \ell_i' \cdot w^3 \sigma'(z_i^2) \cdot \sum_{j \in \mathcal{V}} A_{ij} w^2 \sigma'(z_j^1) \cdot \sum_{j' \in \mathcal{V}} A_{jj'} x_{j'}^0.$$

The nesting of the summations is conspicuous and indicates the neighborhood explosion: when seeking gradients in a $K$-layer GCN on a graph with average degree $m$, up to an $m^{K-1}$-sized neighborhood of a node may be involved in the back-propagation update initiated at that node. Note that the above expression involves terms such as $\sigma'(z_i^2), \sigma'(z_j^1)$. Since the values of $z_i^2, z_j^1$ etc change whenever the model i.e. $\{w^1, w^2, w^3\}$ receives updates, for a fresh mini-batch of nodes, terms such as $\sigma'(z_i^2), \sigma'(z_j^1)$ need to be computed afresh if the gradient is to be computed exactly. Performing these computations amounts to doing *forward pass operations* that frequently involve a large neighborhood of the nodes of the mini-batch. Sampling strategies try to limit this cost by directly restricting the neighborhood over which such forward passes are computed. However, this introduces both bias and variance into the gradient updates as discussed in Section 2. IGLU instead lazily updates various *incomplete gradient* (defined below) and node embedding terms that participate in the gradient expression. This completely eliminates sampling variance but introduces a bias due to the use of stale terms. However, this bias provably bounded and can be made arbitrarily small by adjusting the step length and frequency of refreshing these terms.

**Lazy Updates for GCN Training**: Consider an arbitrary GCN architecture with the following structure: for some parameterized layer functions we have $X^k = f(X^{k-1}; E^k)$ where $E^k$ denotes the collection of all parameters of the $k^{\text{th}}$ layer e.g. weight matrices, offset and scale parameters used in layer norm operations, etc. $X^k \in \mathbb{R}^{N \times d_k}$ denotes the matrix of $k^{\text{th}}$ layer embeddings stacked together and $X^0 \in \mathbb{R}^{N \times d_0}$ are the initial features. For a $K$-layer GCN on a multi-label/multi-class task with $C$ labels/classes, a fully-connected layer $W^{K+1} \in \mathbb{R}^{d_K \times C}$ is used to offer predictions $\hat{\mathbf{y}}_i = (W^{K+1})^\top \mathbf{x}_i^K \in \mathbb{R}^C$. We use the shorthand $\hat{\mathbf{Y}} \in \mathbb{R}^{N \times C}$ to denote the matrix where the predicted outputs $\hat{\mathbf{y}}_i$ for all the nodes are stacked. We assume a task loss function $\ell : \mathbb{R}^C \times \mathbb{R}^C \times \mathbb{R}_+$ and use the abbreviation $\ell_i := \ell(\hat{y}_i, y_i)$. The loss function need not decompose over the classes and can thus be assumed to include activations such as softmax that are applied over the predictions $\hat{\mathbf{y}}_i$. Let $\mathcal{L} = \sum_{i \in \mathcal{V}} \ell_i$ denote the training objective. The convergence proofs assume that $\mathcal{L}$ is smooth.

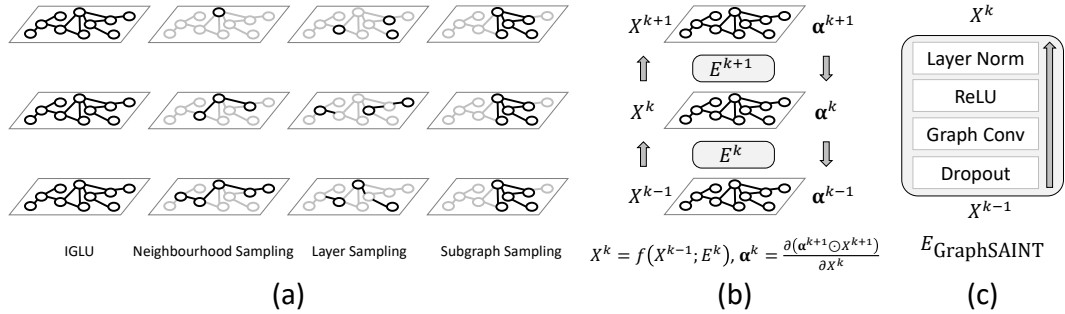

Figure 1: Fig 1(a) highlights the distinction between existing sampling-based approaches that may introduce bias and variance. IGLU completely sidesteps these issues and is able to execute GCN back-propagation steps on the full graph owing to its use of lazy updates which offer no sampling variance and provably bounded bias. Fig 1(b) summarizes the quantities useful for IGLU's updates. IGLU is architecture-agnostic and can be readily used with wide range of architectures. Fig 1(c) gives an example layer architecture used by GraphSAINT.

**Motivation**: We define the the *loss derivative* matrix $\mathbf{G} = [g_{ic}] \in \mathbb{R}^{N \times C}$ with $g_{ic} := \frac{\partial \ell_i}{\partial \hat{y}_{ic}}$. As the proof of Lemma 1 (see Appendix C) shows, the loss derivative with respect to parameters $E^k$ at any layer has the form $\frac{\partial \mathcal{L}}{\partial E^k} = \sum_{j=1}^{N} \sum_{p=1}^{d_k} \left( \sum_{i \in \mathcal{V}} \sum_{c \in [C]} g_{ic} \cdot \frac{\partial \hat{y}_{ic}}{\partial X_{jp}^k} \right) \frac{\partial X_{jp}^k}{\partial E^k}$. Note that the partial derivatives $\frac{\partial X_{jp}^k}{\partial E^k}$ can be computed for any node using only embeddings of its neighbors in the $(k-1)^{\text{th}}$ layer i.e. $X^{k-1}$ thus avoiding any neighborhood explosion. This means that neighborhood explosion must be happening while computing the terms encapsulated in the round brackets. Let us formally recognize these terms as *incomplete gradients*. The notation $\frac{\partial P}{\partial Q}\Big|_R$ denotes the partial derivative of $P$ w.r.t $Q$ while keeping $R$ fixed i.e. treated as a constant.

**Definition 1.** *For any layer $k \leq K$, define its* incomplete task gradient *to be $\boldsymbol{\alpha}^k = [\alpha_{jp}^k] \in \mathbb{R}^{N \times d_k}$,*

$$\alpha_{jp}^k := \frac{\partial(\mathbf{G} \odot \hat{\mathbf{Y}})}{\partial X_{jp}^k}\Bigg|_{\mathbf{G}} = \sum_{i \in \mathcal{V}} \sum_{c \in [C]} g_{ic} \cdot \frac{\partial \hat{y}_{ic}}{\partial X_{jp}^k}$$

The following lemma completely characterizes the loss gradients and also shows that the incomplete gradient terms $\boldsymbol{\alpha}^k, k \in [K]$ can be efficiently computed using a recursive formulation that also does not involve any neighborhood explosion.

**Lemma 1.** *The following results hold whenever the task loss $\mathcal{L}$ is differentiable:*

1. *For the final fully-connected layer we have $\frac{\partial \mathcal{L}}{\partial W^{K+1}} = (X^K)^\top \mathbf{G}$ as well as for any $k \in [K]$ and any parameter $E^k$ in the $k^{th}$ layer, $\frac{\partial \mathcal{L}}{\partial E^k} = \frac{\partial(\boldsymbol{\alpha}^k \odot X^k)}{\partial E^k}\Big|_{\boldsymbol{\alpha}^k} = \sum_{i \in \mathcal{V}} \sum_{p=1}^{d_k} \alpha_{ip}^k \cdot \frac{\partial X_{ip}^k}{\partial E^k}$.*

2. *For the final layer, we have $\boldsymbol{\alpha}^K = \mathbf{G}(W^{K+1})^\top$ as well as for any $k < K$, we have $\boldsymbol{\alpha}^k = \frac{\partial(\boldsymbol{\alpha}^{k+1} \odot X^{k+1})}{\partial X^k}\Big|_{\boldsymbol{\alpha}^{k+1}}$ i.e. $\alpha_{jp}^k = \sum_{i \in \mathcal{V}} \sum_{q=1}^{d_{k+1}} \alpha_{iq}^{k+1} \cdot \frac{\partial X_{iq}^{k+1}}{\partial X_{jp}^k}$.*

Lemma 1 establishes a recursive definition of the incomplete gradients using terms such as $\frac{\partial X_{iq}^{k+1}}{\partial X_{jp}^k}$ that concern just a single layer. Thus, computing $\boldsymbol{\alpha}^k$ for any $k \in [K]$ does not involve any neighborhood explosion since only the immediate neighbors of a node need be consulted. Lemma 1 also shows that if $\boldsymbol{\alpha}^k$ are computed and frozen, the loss derivatives $\frac{\partial \mathcal{L}}{\partial E^k}$ only involve additional computation of terms such $\frac{\partial X_{ip}^k}{\partial E^k}$ which yet again involve a single layer and do not cause neighborhood explosion. This motivates lazy updates to $\boldsymbol{\alpha}^k, X^k$ values in order to accelerate back-propagation. However, performing lazy updates to both $\boldsymbol{\alpha}^k, X^k$ offers suboptimal performance. Hence IGLU adopts two variants described in Algorithms 1 and 2. The *backprop* variant[*] keeps embeddings $X^k$ stale for

---

[*]The backprop variant is named so since it updates model parameters in the order back-propagation would have updated them i.e. $W^{K+1}$ followed by $E^K, E^{K-1}, \dots$ whereas the inverted variant performs updates in the reverse order i.e. starting from $E^1, E^2$ all the way to $W^{K+1}$.

an entire epoch but performs eager updates to $\boldsymbol{\alpha}^k$. The *inverted* variant on the other hand keeps the incomplete gradients $\boldsymbol{\alpha}^k$ stale for an entire epoch but performs eager updates to $X^k$.

| **Algorithm 1** IGLU: backprop order | **Algorithm 2** IGLU: inverted order |
|---|---|
| **Input:** GCN $\mathcal{G}$, initial features $X^0$, task loss $\mathcal{L}$ | **Input:** GCN $\mathcal{G}$, initial features $X^0$, task loss $\mathcal{L}$ |
| 1: Initialize model parameters $E^k, k \in [K], W^{K+1}$ | 1: Initialize model parameters $E^k, k \in [K], W^{K+1}$ |
| 2: **while** not converged **do** | 2: Do an initial forward pass to compute $X^k, k \in [K]$ |
| 3:     Do a forward pass to compute $X^k$ for all $k \in [K]$ as well as $\hat{\mathbf{Y}}$ | 3: **while** not converged **do** |
| 4:     Compute $\mathbf{G}$ then $\frac{\partial \mathcal{L}}{\partial W^{K+1}}$ using Lemma 1 (1) and update $W^{K+1} \leftarrow W^{K+1} - \eta \cdot \frac{\partial \mathcal{L}}{\partial W^{K+1}}$ | 4:     Compute $\hat{\mathbf{Y}}, \mathbf{G}$ and $\boldsymbol{\alpha}^k$ for all $k \in [K]$ using Lemma 1 (2) |
| 5:     Compute $\boldsymbol{\alpha}^K$ using $\mathbf{G}, W^{K+1}$, Lemma 1 (2) | 5:     **for** $k = 1 \ldots K$ **do** |
| 6:     **for** $k = K \ldots 2$ **do** | 6:         Compute $\frac{\partial \mathcal{L}}{\partial E^k}$ using $\boldsymbol{\alpha}^k, X^k$, Lemma 1 (1) |
| 7:         Compute $\frac{\partial \mathcal{L}}{\partial E^k}$ using $\boldsymbol{\alpha}^k, X^k$, Lemma 1 (1) | 7:         Update $E^k \leftarrow E^k - \eta \cdot \frac{\partial \mathcal{L}}{\partial E^k}$ |
| 8:         Update $E^k \leftarrow E^k - \eta \cdot \frac{\partial \mathcal{L}}{\partial E^k}$ | 8:         Update $X^k \leftarrow f(X^{k-1}; E^k)$ |
| 9:         Update $\boldsymbol{\alpha}^k$ using $\boldsymbol{\alpha}^{k+1}$ using Lemma 1 (2) | 9:     **end for** |
| 10:     **end for** | 10:     Compute $\frac{\partial \mathcal{L}}{\partial W^{K+1}}$ using Lemma 1 (1) and use it to update $W^{K+1} \leftarrow W^{K+1} - \eta \cdot \frac{\partial \mathcal{L}}{\partial W^{K+1}}$ |
| 11: **end while** | 11: **end while** |

**SGD Implementation**: Update steps in the algorithms (steps 4, 8 in Algorithm 1 and steps 7, 10 in Algorithm 2) are described as a single gradient step over the entire graph to simplify exposition – in practice, these steps are implemented using *mini-batch* SGD. A mini-batch of nodes $S$ is sampled and task gradients are computed w.r.t $\hat{\mathcal{L}}_S = \sum_{i \in S} \ell_i$ alone instead of $\mathcal{L}$.

**Contribution**: As noted in Section 2, IGLU uses caching in a manner fundamentally different from frameworks such as PyTorch or TensorFlow which use short-lived caches and compute exact gradients unlike IGLU that computes gradients faster but with bounded bias. Moreover, unlike techniques such as VR-GCN that cache only node embeddings, IGLU instead offers two variants and the variant that uses inverted order of updates (Algorithm 2) and caches incomplete gradients usually outperforms the backprop variant of IGLU (Algorithm 1) that caches node embeddings.

**Theoretical Analysis**: Conditioned on the stale parameters (either $\boldsymbol{\alpha}^k$ or $X^k$ depending on which variant is being executed), the gradients used by IGLU to perform model updates (steps 4, 8 in Algorithm 1 and steps 7, 10 in Algorithm 2) do not have any sampling bias. However, the staleness itself training bias. However, by controlling the step length $\eta$ and the frequency with which the stale parameters are updated, this bias can be provably controlled resulting in guaranteed convergence to a first-order stationary point. The detailed statement and proof are presented in Appendix C.

**Theorem 2** (IGLU Convergence (Informal)). *Suppose the task objective $\mathcal{L}$ has $\mathcal{O}(1)$-Lipschitz gradients and IGLU is executed with small enough step lengths $\eta$ for model updates (steps 4, 8 in Algorithm 1 and steps 7, 10 in Algorithm 2), then within $T$ iterations, IGLU ensures:*

*1. $\|\nabla \mathcal{L}\|_2^2 \le \mathcal{O}\left(1/T^{\frac{2}{3}}\right)$ if update steps are carried out on the entire graph in a full-batch.*

*2. $\|\nabla \mathcal{L}\|_2^2 \le \mathcal{O}\left(1/\sqrt{T}\right)$ if update steps are carried out using mini-batch SGD.*

This result holds under minimal assumptions of objective smoothness and boundedness that are standard Chen et al. (2018a); Cong et al. (2020), yet offers convergence rates comparable to those offered by standard mini-batch SGD. However, whereas works such as (Chen et al., 2018a) assume bounds on the sup-norm i.e. $L_\infty$ norm of the gradients, Theorem 2 only requires an $L_2$ norm bound. Note that objective smoothness requires the architecture to use smooth activation functions. However, IGLU offers similar performance whether using non-smooth activations e.g. ReLU or smooth ones e.g. GELU (see Appendix B.7) as is also observed by other works (Hendrycks & Gimpel, 2020).

## 4 EMPIRICAL EVALUATION

IGLU was compared to state-of-the-art (SOTA) baselines on several node classification benchmarks in terms of test accuracy and convergence rate. The inverted order of updates was used for IGLU as it was found to offer superior performance in ablation studies.

**Datasets and Tasks:** The following five benchmark tasks were used:
(1) Reddit (Hamilton et al., 2017): predicting the communities to which different posts belong,
(2) PPI-Large (Hamilton et al., 2017): classifying protein functions in biological protein-protein interaction graphs,
(3) Flickr (Zeng et al., 2020): image categorization based on descriptions and other properties,
(4) OGBN-Arxiv (Hu et al., 2020): predicting paper-paper associations, and
(5) OGBN-Proteins (Hu et al., 2020): categorizing meaningful associations between proteins.
Training-validation-test splits and metrics were used in a manner consistent with the original release of the datasets: specifically ROC-AUC was used for OGBN-Proteins and micro-F1 for all other datasets. Dataset descriptions and statistics are presented in Appendix B. The graphs in these tasks varied significantly in terms of size (from 56K nodes in PPI-Large to 232K nodes in Reddit), density (from average degree 13 in OGBN-Arxiv to 597 in OGBN-Proteins) and number of edges (from 800K to 39 Million). They require diverse information to be captured and a variety of multi-label and multi-class node classification problems to be solved thus offering extensive evaluation.

**Baselines:** IGLU was compared to state-of-the-art algorithms, namely - GCN (Kipf & Welling, 2017), GraphSAGE (Hamilton et al., 2017), VR-GCN (Chen et al., 2018a), Cluster-GCN (Chiang et al., 2019) and GraphSAINT (Zeng et al., 2020) (using the Random Walk Sampler which was reported by the authors to have the best performance). The mini-batched implementation of GCN provided by GraphSAGE authors was used since the implementation released by (Kipf & Welling, 2017) gave run time errors on all datasets. GraphSAGE and VRGCN address the neighborhood explosion problem by sampling neighborhood subsets, whereas ClusterGCN and GraphSAINT are subgraph sampling techniques. Thus, our baselines include neighbor sampling, layer-wise sampling, subgraph sampling and no-sampling methods. We recall that IGLU does not require any node/subgraph sampling. IGLU was implemented in TensorFlow and compared with TensorFlow implementations of the baselines released by the authors. Due to lack of space, comparisons with the following additional baselines is provided in Appendix B.2: LADIES (Zou et al., 2019), L2-GCN (You et al., 2020), AS-GCN (Huang et al., 2018), MVS-GNN (Cong et al., 2020), FastGCN (Chen et al., 2018b), SIGN (Frasca et al., 2020), PPRGo (Bojchevski et al., 2020) and Bandit Sampler (Liu et al., 2020).

**Architecture**: We note that all baseline methods propose a specific network architecture along with their proposed training strategy. These architectures augment the standard GCN architecture (Kipf & Welling, 2017) e.g. using multiple non-linear layers within each GCN layer, normalization and concatenation layers, all of which can help improve performance. IGLU being architecture-agnostic can be readily used with all these architectures. However, for the experiments, the network architectures proposed by VR-GCN and GraphSAINT was used with IGLU owing to their consistent performance across datasets as demonstrated by (Chen et al., 2018a; Zeng et al., 2020). Both architectures were considered and results for the best architecture are reported for each dataset.

**Detailed Setup.** The supervised inductive learning setting was considered for all five datasets as it is more general than the transductive setting that assumes availability of the entire graph during training. The inductive setting also aligns with real-world applications where graphs can grow over time (Hamilton et al., 2017). Experiments on PPI-Large, Reddit and Flickr used 2 Layer GCNs for all methods whereas OGBN-Proteins and OGBN-Arxiv used 3 Layer GCNs for all methods as prescribed in the original benchmark (Hu et al., 2020). Further details are provided in Appendix A.1.

**Model Selection and Hyperparameter Tuning.** Model selection was done for all methods based on their validation set performance. Each experiment was run five times with mean and standard deviation in test performance reported in Table 1 along with training times. Although the embedding dimension varies across datasets, they are same for all methods for any dataset. For IGLU, GraphSAGE and VR-GCN, an exhaustive grid search was done over general hyperparameters such as batch size, learning rate and dropout rate (Srivastava et al., 2014). In addition, method-specific hyperparameter sweeps were also carried out that are detailed in Appendix A.4.

## 4.1 RESULTS

For IGLU, the VR-GCN architecture performed better for Reddit and OGBN-Arxiv datasets while the GraphSAINT architecture performed better on the remaining three datasets. All baselines were extensively tuned on the 5 datasets and their performance reported in their respective publications was either replicated closely or else improved. Test accuracies are reported in Table 1 and convergence

Table 1: **Accuracy of IGLU compared to SOTA algorithms.** The metric is ROC-AUC for Proteins and Micro-F1 for the others. IGLU is the only method with accuracy within $0.2\%$ of the best accuracy on each dataset with significant speedups in training across datasets. On PPI-Large, IGLU is $\sim 8\times$ faster than VR-GCN, the most accurate baseline. * denotes speedup in initial convergence based on a high validation score of 0.955. – denotes no absolute gain. ‖ denotes a runtime error.

| Algorithm | PPI-Large | Reddit | Flickr | Proteins | Arxiv |
|---|---|---|---|---|---|
| GCN | $0.614 \pm 0.004$ | $0.931 \pm 0.001$ | $0.493 \pm 0.002$ | $0.615 \pm 0.004$ | $0.657 \pm 0.002$ |
| GraphSAGE | $0.736 \pm 0.006$ | $0.954 \pm 0.002$ | $0.501 \pm 0.013$ | $0.759 \pm 0.008$ | $0.682 \pm 0.002$ |
| VR-GCN | $0.975 \pm 0.007$ | $0.964 \pm 0.001$ | $0.483 \pm 0.002$ | $0.752 \pm 0.002$ | $0.701 \pm 0.006$ |
| Cluster-GCN | $0.899 \pm 0.004$ | $0.962 \pm 0.004$ | $0.481 \pm 0.005$ | ‖ | $0.706 \pm 0.004$ |
| GraphSAINT | $0.956 \pm 0.003$ | $\mathbf{0.966} \pm 0.003$ | $0.510 \pm 0.001$ | $0.764 \pm 0.009$ | $0.712 \pm 0.006$ |
| IGLU | $\mathbf{0.987} \pm 0.004$ | $0.964 \pm 0.001$ | $\mathbf{0.515} \pm 0.001$ | $\mathbf{0.784} \pm 0.004$ | $\mathbf{0.718} \pm 0.001$ |
| Abs. Gain | **0.012** | **–** | **0.005** | **0.020** | **0.006** |
| % Speedup (1) | **88.12** | **8.1**\* | **44.74** | **11.05** | **13.94** |

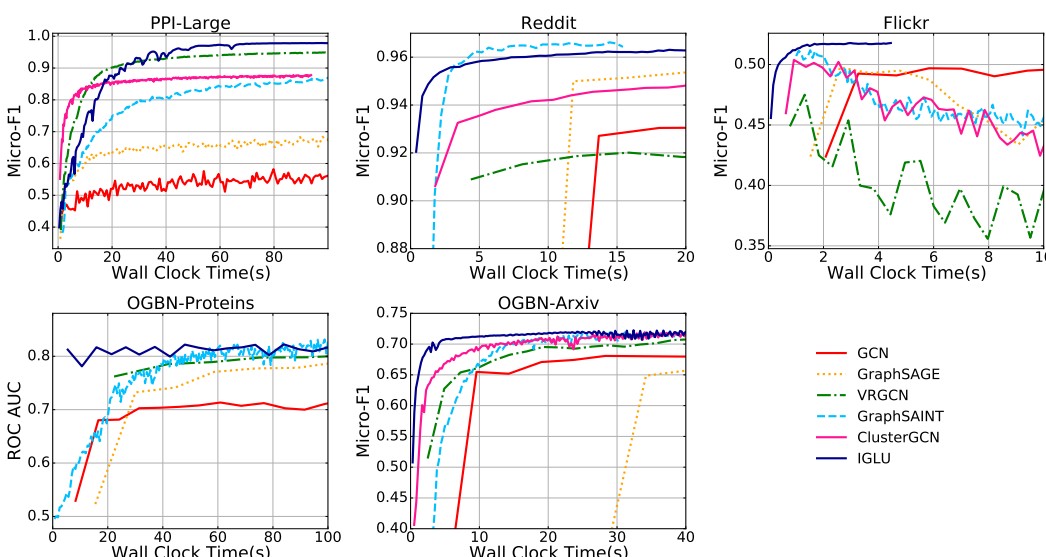

Figure 2: **Wall Clock Time vs Validation Accuracy on different datasets for various methods.** IGLU offers significant improvements in convergence rate over baselines across diverse datasets.

plots are shown in Figure 2. Additionally, Table 1 also reports the absolute accuracy gain of IGLU over the best baseline. The % speedup offered by IGLU is computed as follows: let the highest validation score obtained by the best baseline be $v_1$ and $t_1$ be the time taken to reach that score. Let the time taken by IGLU to reach $v_1$ be $t_2$. Then,

$$\% \text{ Speedup} := \frac{t_1 - t_2}{t_1} \times 100 \qquad (1)$$

The wall clock training time in Figure 2 is strictly the optimization time for each method and excludes method-specific overheads such as pre-processing, sampling and sub-graph creation that other baselines incur. This is actually a disadvantage to IGLU since its overheads are much smaller. The various time and memory overheads incurred by all methods are summarized in Appendix A.2 and memory requirements for IGLU are discussed in Appendix A.3.

**Performance on Test Set and Speedup Obtained:** Table 1 establishes that IGLU significantly outperforms the baselines on PPI-Large, Flickr, OGBN-Proteins and OGBN-Arxiv and is competitive with best baselines on Reddit. On PPI-Large, IGLU improves accuracy upon the best baseline (VRGCN) by over **1.2**% while providing speedups of upto **88**%, i.e., IGLU is about $8\times$ faster to train than VR-GCN. On OGBN-Proteins, IGLU improves accuracy upon the best baseline (GraphSAINT) by over **2.6**% while providing speedup of **11**%. On Flickr, IGLU offers **0.98**% improvement in

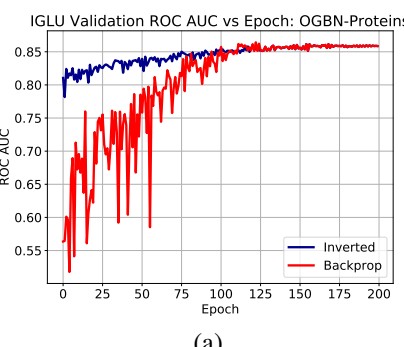 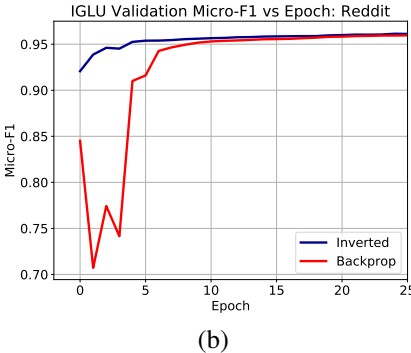

|(a)|(b)|

Figure 3: Effect of backprop and inverted order of updates in IGLU on Reddit and OGBN-Proteins datasets. The inverted order of updates offers more stability and faster convergence. It is notable that techniques such as VR-GCN use stale node embeddings that correspond to the backprop variant.

accuracy while simultaneously offering upto **45**% speedup over the previous state-of-the-art method GraphSAINT. Similarly on Arxiv, IGLU provides **0.84**% accuracy improvement over the best baseline GraphSAINT while offering nearly **14**% speedup. On Reddit, an **8.1**% speedup was observed in convergence to a high validation score of 0.955 while the final performance is within a standard deviation of the best baseline. The OGBN-Proteins and OGBN-Arxiv datasets were originally benchmarked in the transductive setting, with the entire graph information made available during training. However, we consider the more challenging inductive setting yet IGLU outperforms the best transductive baseline by over **0.7**% for Proteins while matching the best transductive baseline for Arxiv (Hu et al., 2020). It is important to note that OGBN-Proteins is an atypical dataset because the graph is highly dense. Because of this, baselines such as ClusterGCN and GraphSAINT that drop a lot of edges while creating subgraphs show a deterioration in performance. ClusterGCN encounters into a runtime error on this dataset (denoted by ∥ in the table), while GraphSAINT requires a very large subgraph size to achieve reasonable performance.

**Convergence Analysis**: Figure 2 shows that IGLU converges faster to a higher validation score than other baselines. For PPI-Large, while Cluster-GCN and VR-GCN show promising convergence in the initial stages of training, they stagnate at a much lower validation score in the end whereas IGLU is able to improve consistently and converge to a much higher validation score. For Reddit, IGLU's final validation score is marginally lower than GraphSAINT but IGLU offers rapid convergence in the early stages of training. For OGBN-Proteins, Flickr and OGBN-Arxiv, IGLU demonstrates a substantial improvement in both convergence and the final performance on the test set.

## 4.2 ABLATION STUDIES

**Effect of the Order of Updates.** IGLU offers the flexibility of using either the backprop order of updates or the inverted order of updates as mentioned in Section 3. Ablations were performed on the Reddit and OGBN-Proteins datasets to analyse the effect of the different orders of updates. Figure 3 offers epoch-wise convergence for the same and shows that the inverted order of updates offers faster and more stable convergence in the early stages of training although both variants eventually converge to similar validation scores. It is notable that keeping node embeddings stale (backprop order) offered inferior performance since techniques such as VR-GCN (Chen et al., 2018a) that also keep node embeddings stale. IGLU offers the superior alternative of keeping incomplete gradients stale instead.

**Analysis of Degrees of Staleness.** The effect of frequency of updates to the incomplete gradients ($\alpha^k$) on the performance and convergence of IGLU was analyzed. This ablation was conducted keeping all the other hyperparameters fixed. In the default setting, $\alpha^k$ values were updated only once per epoch (referred to as frequency 1). Two other update schemes were also considered: a) update the $\alpha^k$ values every two epochs (referred to as frequency 0.5), and b) update the $\alpha^k$ values twice within an epoch (referred to as frequency 2). To clarify, $\alpha^k$ values are the most fresh with update frequency 2 and most stale with update frequency 0.5. This ablation study was performed on the PPI-Large dataset and each variant was trained for 200 epochs. Table 2 summarizes the results doing model selection and Figure 4 plots the convergence of these variants. Figure 4b shows that on PPI-Large, the default

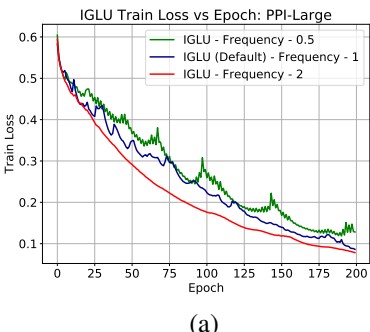 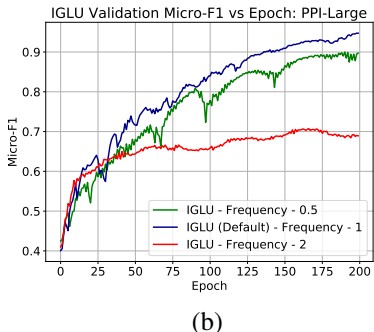

(a)                (b)

Figure 4: Update frequency vs Accuracy. Experiments conducted on PPI-Large. As expected, refreshing the $\boldsymbol{\alpha}^k$'s too frequently or too infrequently can affect both performance and convergence.

update frequency 1 has the best convergence on the validation set, followed by update frequency 0.5. Both update frequency 1 and 0.5 massively outperform update frequency 2. Figure 4a shows that IGLU with update frequency 2 has the lowest training loss but poor validation performance, indicating overfitting to the training dataset. We observe from this ablation that updating embeddings prematurely can cause unstable training resulting in convergence to a suboptimal solution. However, not refreshing the $\boldsymbol{\alpha}^k$ values frequently enough can delay convergence to a good solution.

Table 2: Accuracy vs different update frequency on PPI-Large.

| Update Frequency | Train Micro-F1 | Validation Micro-F1 | Test Micro-F1 |
|---|---|---|---|
| 0.5 | 0.947 | 0.899 | 0.916 |
| 1 | 0.970 | 0.947 | 0.961 |
| 2 | 0.960 | 0.708 | 0.726 |

**Additional Ablations and Experiments:** Due to lack of space, the following additional ablations and experiments are described in the appendices: a) Ablation on degrees of staleness for the backprop order of updates in Appendix B.8, b) Experiments demonstrating IGLU's scalability to deeper networks and larger datasets in Appendices B.3 and B.4 respectively, and c) Experiments demonstrating IGLU's applicability and architecture-agnostic nature in Appendices B.5 and B.6.

## 5 DISCUSSION AND FUTURE WORK

This paper introduced IGLU, a novel method for training GCN architectures that uses biased gradients based on cached intermediate computations to speed up training significantly. The gradient bias is shown to be provably bounded so overall convergence is still effective (see Theorem 2).IGLU's performance was validated on several datasets where it significantly outperformed SOTA methods in terms of accuracy and convergence speed. Ablation studies confirmed that IGLU is robust to its few hyperparameters enabling a near-optimal choice. Exploring other possible variants of IGLU, in particular reducing variance due to mini-batch SGD, sampling nodes to further speed-up updates, and exploring alternate staleness schedules are interesting future directions. On a theoretical side, it would be interesting to characterize properties of datasets and loss functions that influence the effect of lazy updates on convergence. Having such a property would allow practitioners to decide whether to execute IGLU with lazier updates or else reduce the amount of staleness. Exploring application- and architecture-specific variants of IGLU is also an interesting direction.

### REPRODUCIBILITY STATEMENT

Efforts have been made to ensure that results reported in this paper are reproducible.

**Theoretical Clarity**: Section 3 discusses the problem setup and preliminaries and describes the proposed algorithm. Detailed proofs are provided in Appendix C due to lack of space.

**Experimental Reproducibility**: Section 4 and Appendices A and B contain information needed to reproduce the empirical results, namely datasets statistics and data source, data pre-processing, implementation details for IGLU and the baselines including architectures, hyperparameter search spaces and the best hyperparameters corresponding to the results reported in the paper.

**Code Release**: An implementation of IGLU can be found at the following URL
`https://github.com/sdeepaknarayanan/iglu`

ETHICS STATEMENT

This paper presents IGLU, a novel technique to train GCN architectures on large graphs that outperforms state of the art techniques in terms of prediction accuracy and convergence speed. The paper does not explore any sensitive applications and the experiments focus primarily on publicly available benchmarks of scientific (e.g. PPI) and bibliographic (e.g. ArXiv) nature do not involve any user studies or human experimentation.

ACKNOWLEDGMENTS

The authors are thankful to the reviewers for discussions that helped improve the content and presentation of the paper.

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

## APPENDIX

This appendix is segmented into three key parts.

1. **Section A** discusses additional implementation details. In particular, method-specific overheads are discussed in detail and detailed hyper-parameter settings for IGLU and the main baselines reported in Table 1 are provided. A key outcome of this analysis is that IGLU has the least overheads as compared to the other methods. We also provide the methodology for incorporating architectural modifications into IGLU, provide a detailed comparison with caching based methods and provide a more detailed descriptions of the algorithms mentioned in the main paper.

2. **Section B** reports dataset statistics, provides comparisons with additional baselines (not included in the main paper due to lack of space) and provides experiments on scaling to deeper models and larger graphs as mentioned in Section 4.2 in the main paper. It also provides experiments for the applicability of IGLU across settings and architectures, using smooth activation functions, continued ablation on degrees of staleness and optimization on the train set. The key outcome of this discussion is that IGLU scales well to deeper models and larger datasets, and continues to give performance boosts with state-of-the-art even when compared to these additional baselines. Experimental evidence also demonstrates IGLU's ability to generalize easily to a different setting and architecture, perform equivalently with smooth activation functions and achieve lower training losses faster compared to all the baselines across datasets.

3. **Section C** gives detailed proofs of Lemma 1 and the convergence rates offered by IGLU. It is shown that under standard assumptions such as objective smoothness, IGLU is able to offer both the standard rate of convergence common for SGD-style algorithms, as well as a *fast* rate if full-batch GD is performed.

## A    ADDITIONAL IMPLEMENTATION DETAILS

We recall from Section 4.1 that the wall-clock time reported in Figure 2 consists of strictly the optimization time for each method and excludes method-specific overheads. This was actually a disadvantage for IGLU since its overheads are relatively mild. This section demonstrates that other baselines incur much more significant overheads whereas IGLU does not suffer from these large

overheads. When included, it further improves the speedups that IGLU provides over baseline methods.

## A.1 HARDWARE

We implement IGLU in TensorFlow 1.15.2 and perform all experiments on an NVIDIA V100 GPU (32 GB Memory) and Intel Xeon CPU processor (2.6 GHz). We ensure that all baselines are experimented with the exact same hardware.

## A.2 TIME AND MEMORY OVERHEADS FOR VARIOUS METHODS

We consider three main types of overheads that are incurred by different methods. This includes pre-processing overhead that is one-time, recurring overheads and additional memory overhead. We describe each of the overheads in the context of the respective methods below.

1. **GraphSAGE** - GraphSAGE recursively samples neighbors at each layer for every minibatch. This is done on-the-fly and contributes to a significant sampling overhead. Since this overhead is incurred for every minibatch, we categorize this under recurring overhead. We aggregate this overhead across all minibatches during training. GraphSAGE does not incur preprocessing or additional memory overheads.

2. **VRGCN** - Similar to GraphSAGE, VRGCN also recursively samples neighbors at each layer for every minibatch on-the-fly. We again aggregate this overhead across all minibatches during training. VRGCN also stores the stale/historical embeddings that are learnt for every node at every layer. This is an additional overhead of $\mathcal{O}\left(NKd\right)$, where $K$ is the number of layers, $N$ is the number of nodes in the graph and $d = \frac{1}{K}\sum_{k\in[K]} d_k$ is the average embedding dimensionality across layers.

3. **ClusterGCN** - ClusterGCN creates subgraphs and uses them as minibatches for training. For the creation of these subgraphs, ClusterGCN performs graph clustering using the highly optimized METIS tool[†]. This overhead is a one-time overhead since graph clustering is done before training and the same subgraphs are (re-)used during the whole training process. We categorize this under preprocessing overhead. ClusterGCN does not incur any recurring or additional memory overheads.

4. **GraphSAINT** - GraphSAINT, similar to ClusterGCN creates subgraphs to be used as mini-batches for training. We categorize this minibatch creation as the preprocessing overhead for GraphSAINT. However, unlike ClusterGCN, GraphSAINT also periodically creates new subgraphs on-the-fly. We categorize this overhead incurred in creating new subgraphs as recurring overhead. GraphSAINT does not incur any additional memory overheads.

5. **IGLU** - IGLU creates mini-batches only once using subsets of nodes with their full neighborhood information which is then reused throughout the training process. In addition to this, IGLU requires initial values of both the incomplete gradients $\boldsymbol{\alpha}^k$ and the $X^k$ embeddings (Step 3 and first part of Step 4 in Algorithm 1 and Step 2 and Step 4 in Algorithm 2) before optimization can commence. We categorize these two overheads - mini-batch creation and initializations of $\boldsymbol{\alpha}^k$'s and $X^k$ embeddings as IGLU's preprocessing overhead and note that **IGLU does not have any recurring overheads**. IGLU does incur an additional memory overhead since it needs to store the incomplete gradients $\boldsymbol{\alpha}^k$'s in the inverted variant and the embeddings $X^k$ in the backprop variant. However, note that the memory occupied by $X^k$ for a layer $k$ is the same as that occupied by $\boldsymbol{\alpha}^k$ for that layer (see Definition 1). Thus, for both its variants, IGLU incurs an additional memory overhead of $\mathcal{O}\left(NKd\right)$, where $K$ is the number of layers, $N$ is the number of nodes in the graph and $d = \frac{1}{K}\sum_{k\in[K]} d_k$ is the average embedding dimensionality across layers.

Tables 3 and 4 report the overheads incurred by different methods on the OGBN-Proteins and Reddit datasets (the largest datasets in terms of edges and nodes respectively). ClusterGCN runs into a runtime error on the Proteins dataset **(denoted by ‖ in the table)**. N/A stands for Not Applicable in the tables. In the tables, specifically for IGLU, pre-processing time is the sum of initialization time

---

[†]http://glaros.dtc.umn.edu/gkhome/metis/metis/download

Table 3: **OGBN-Proteins: Overheads of different methods.** IGLU does not have any recurring overhead while VRGCN, GraphSAGE and GraphSAINT all suffer from heavy recurring overheads. ClusterGCN runs into runtime error on this dataset (**denoted by ‖**). GraphSAINT incurs an overhead that is $\sim 2\times$ the overhead incurred by IGLU, while GraphSAGE and VRGCN incur upto $\sim 4.7\times$ and $\sim 7.8\times$ the overhead incurred by IGLU respectively. For the last row, I denotes initialization time, MB denotes minibatch time and T denotes total preprocessing time. Please refer to Discussion on OGBN - Proteins at Section A.2 for more details.

| Method | Preprocessing (One-time) | Recurring | Additional Memory |
|---|---|---|---|
| GraphSAGE | N/A | 276.8s | N/A |
| VRGCN | N/A | 465.0s | $\mathcal{O}\left(NKd\right)$ |
| ClusterGCN | ‖ | ‖ | ‖ |
| GraphSAINT | 22.1s | 101.0s | N/A |
| IGLU | 34.0s (I) + 25.0s (MB) = 59.0s (T) | N/A | $\mathcal{O}\left(NKd\right)$ |

Table 4: **Reddit: Overheads of different methods.** IGLU and GraphSAINT do not have any recurring overhead for this dataset while VRGCN and GraphSAGE incur heavy recurring overheads. ClusterGCN suffers from heavy preprocessing overhead incurred due to clustering. In this case, IGLU incurs an overhead that is marginally higher ($\sim 1.4\times$) than that of GraphSAINT, while VRGCN, GraphSAGE and ClusterGCN incur as much as $\sim 2.1\times$, $\sim 4.5\times$ and $\sim 5.8\times$ the overhead incurred by IGLU respectively. For the last line, I denotes initialization time, MB denotes minibatch time and T denotes total preprocessing time. Please refer to Discussion on Reddit at Section A.2 for more details.

| Method | Preprocessing (One-time) | Recurring | Additional Memory |
|---|---|---|---|
| GraphSAGE | N/A | 41.7s | N/A |
| VRGCN | N/A | 19.2s | $\mathcal{O}\left(NKd\right)$ |
| ClusterGCN | 54.0s | N/A | N/A |
| GraphSAINT | 6.7s | N/A | N/A |
| IGLU | 3.5s (I) + 5.7s (MB)= 9.2s (T) | N/A | $\mathcal{O}\left(NKd\right)$ |

required to pre-compute $\boldsymbol{\alpha}^k, X^k$ and mini-batch creation time. We also report the individual overhead for both initialization and mini-batch creation for IGLU. The total pre-processing time for IGLU is denoted by T, overheads incurred for initialization by I and overheads incurred for mini-batch creation by MB.

For IGLU, the minibatch creation code is currently implemented in Python, while GraphSAINT uses a highly optimized C++ implementation. Specifically for the Reddit dataset, the number of subgraphs that GraphSAINT samples initially is sufficient and it does not incur any recurring overhead. However, on Proteins, GraphSAINT samples 200 subgraphs every 18 epochs, once the initially sampled subgraphs are used, leading to a sizeable recurring overhead.

**Discussion on OGBN - Proteins:** Figure 5 summarizes the total overheads for all the methods. On the OGBN-Proteins dataset, IGLU incurs $\sim 2\times$ less overhead than GraphSAINT, the best baseline, while incurring as much as $\sim 7.8\times$ less overhead than VRGCN and $\sim 4.7\times$ less overhead than GraphSAGE. It is also important to note that these overheads are often quite significant as compared to the optimization time for the methods and can add to the overall experimentation time. For experiments on the OGBN-Proteins dataset, VRGCN's total overheads equal 46.25% of its optimization time, GraphSAINT's overheads equal 19.63% of its optimization time, GraphSAGE's overhead equal 9.52% of its optimization time. However IGLU's total overheads equal only **5.59%** of its optimization time which is the lowest out of all methods. Upon re-computing the speedup provided by IGLU using the formula defined in equation (1), but this time with overheads included, it was observed that IGLU offered an improved speedup of **16.88%** (the speedup was only 11.05% when considering only optimization time).

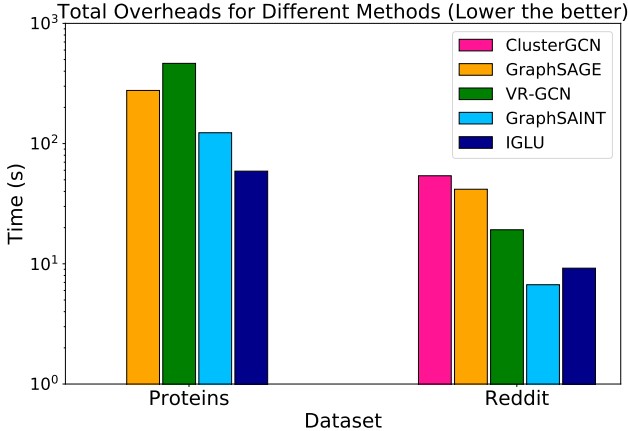

Figure 5: **Total Overheads in Wall Clock Time (Log Scale) for the different methods on OGBN-Proteins and Reddit dataset.** ClusterGCN runs into runtime error on the OGBN-Proteins dataset and hence has not been included in the plot. IGLU frequently offers least total overhead compared to the other methods and hence significantly lower overall experimentation time. Please refer to section A.2 for details.

Table 5: **URL's and commit numbers to run baseline codes**

| Method | URL | Commit |
|---|---|---|
| GCN | github.com/williamleif/GraphSAGE | a0fdef |
| GraphSAGE | github.com/williamleif/GraphSAGE | a0fdef |
| VRGCN | github.com/thu-ml/stochastic_gcn | da7b78 |
| ClusterGCN | github.com/google-research/google-research/tree/master/cluster_gcn | 0c1bbe5 |
| AS-GCN | github.com/huangwb/AS-GCN | 5436ecd |
| L2-GCN | github.com/VITA-Group/L2-GCN | 687fbae |
| MVS-GNN | github.com/CongWeilin/mvs_gcn | a29c2c5 |
| LADIES | github.com/acbull/LADIES | c10b526 |
| FastGCN | https://github.com/matenure/FastGCN | b8e6e64 |
| SIGN | https://github.com/twitter-research/sign | 42a230c |
| PPRGo | https://github.com/TUM-DAML/pprgo_pytorch | c92c32e |
| BanditSampler | https://github.com/xavierzw/gnn-bs | a2415a9 |

**Discussion on Reddit:** On the Reddit dataset, IGLU incurs upto $\sim 2.1\times$ less overhead than VRGCN and upto $\sim 4.5\times$ less overhead than GraphSAGE. However, IGLU incurs marginally higher overhead ($\sim 1.4\times$) than GraphSAINT. This can be attributed to the non-optimized minibatch creation routine currently used by IGLU compared to a highly optimized and parallelized implementation in C++ used by GraphSAINT. This is an immediate avenue for future work. Nevertheless, VRGCN's total overhead equals 15.41% of its optimization time, GraphSAINT's overhead equals 43.41% of its optimization time, GraphSAGE's overhead equals 31.25% of its optimization time, ClusterGCN's overhead equals 41.02% of its optimization time while IGLU's overhead equals 44.09% of its optimization time. Whereas the relative overheads incurred by IGLU and GraphSAINT in comparison to optimization time may seem high for this dataset, this is because the actual optimization times for these methods are rather small, being just 15.43 and 20.86 seconds for GraphSAINT and IGLU respectively in comparison to the other methods such as VRGCN, ClusterGCN and GraphSAGE whose optimization times are 124s, 131s and 133s respectively, almost an order of magnitude larger than that of IGLU.

### A.3 MEMORY ANALYSIS FOR IGLU

While IGLU requires storing stale variables which can have additional memory costs, for most scenarios with real world graphs, saving these stale representations on modern GPUs are quite reasonable. We provide examples of additional memory usage required for two of the large datasets

- Reddit and Proteins in Table 6 and we observe that IGLU requires only **150MB** and **260MB** of additional GPU memory. Even for a graph with 1 million nodes, the additional memory required would only be $\sim 2.86$GB which easily fit on modern GPUs. For even larger graphs, CPU-GPU interfacing can be used. CPU and GPU interfacing for data movement is a common practice in training machine learning models and hence a potential method to mitigate the issue of limited memory availability in settings with large datasets. This has been explored by many works for dealing with large datasets in the context of GCNs, such as VR-GCN (Chen et al., 2018a) for storing historical activations in main memory (CPU). Such an interfacing is an immediate avenue of future work for IGLU.

Table 6: **Additional Memory Overheads incurred by IGLU on Large datasets**

| Dataset | # of Train Nodes | Embedding Dimensions | Number of Layers | Memory Per Layer | Total GPU Memory |
|---|---|---|---|---|---|
| Reddit | 155k | 128 | 2 | 75MB | 150MB |
| Proteins | 90k | 256 | 3 | 85MB | 260MB |
| Million Sized Graph | 1M | 256 | 3 | 0.95GB | 2.86GB |

We observe that IGLU enjoys significant speedups and improvements in training cost across datasets as compared to the baselines as a result of using stale variables. Additionally, since IGLU requires training just a single layer at a time, there is scope for further reduction in memory usage by using only the variables required for the current layer and by re-using the computation graphs across layers, and therefore making IGLU even less memory expensive.

### A.4 HYPERPARAMETER CONFIGURATIONS FOR IGLU AND BASELINES

Table 5 summarizes the source URLs and commit stamps using which baseline methods were obtained for experiments. The Adam optimizer Kingma & Ba (2015) was used to train IGLU and all the baselines until convergence for each of the datasets. A grid search was performed over the other hyper-parameters for each baseline which are summarized below:

1. **GraphSAGE**: Learning Rate - {0.01, 0.001}, Batch Size - {512, 1024, 2048}, Neighborhood Sampling Size - {25, 10, 5}, Aggregator - {Mean, Concat}

2. **VRGCN** : Batch Size - {512, 1000, 2048}, Degree - {1, 5, 10}, Method - {CV, CVD}, Dropout - {0, 0.2, 0.5}

3. **ClusterGCN** : Learning Rate - {0.01, 0.001, 0.005}, Lambda - {-1, 1, 1e-4}, Number of Clusters - {5, 50, 500, 1000, 1500, 5000}, Batch Size - {1, 5, 50, 100, 500}, Dropout - {0, 0.1, 0.3, 0.5}

4. **GraphSAINT-RW** : Aggregator - {Mean, Concat}, Normalization - {Norm, Bias}, Depth - {2, 3, 4}, Root Nodes - {1250, 2000, 3000, 4500, 6000}, Dropout - {0.0, 0.2, 0.5}

5. **IGLU** : Learning Rate - {0.01, 0.001} with learning rate decay schemes, Batch Size - {512, 2048, 4096, 10000}, Dropout - {0.0, 0.2, 0.5, 0.7}

### A.5 INCORPORATING RESIDUAL CONNECTIONS, BATCH NORMALIZATION AND VIRTUAL NODES IN IGLU

General-purpose techniques such as BatchNorm and skip/residual connections, and GCN-specific advancements such as bi-level aggregation using virtual nodes offer performance boosts. The current implementation of IGLU already incorporates normalizations as described in Sections 3 and 4. Below we demonstrate how all these aforementioned architectural variations can be incorporated into IGLU with minimal changes to Lemma 1 part 2. Remaining guarantees such as those offered by Lemma 1 part 1 and Theorem 2 remain unaltered but for changes to constants.

**Incorporating BatchNorm into IGLU**: As pointed out in Section 3, IGLU assumes a general form of the architecture, specifically

$$\mathbf{x}_i^k = f(\mathbf{x}_j^{k-1}, j \in \{i\} \cup \mathcal{N}(i); E^k),$$

where $f$ includes the aggregation operation such as using the graph Laplacian, weight matrices and any non-linearity. This naturally allows operations such as normalizations (LayerNorm, BatchNorm

etc) to be carried out. For instance, the $f$ function for BatchNorm with a standard GCN would look like the following

$$\mathbf{z}_i^k = \sigma \left( \sum_{j \in \mathcal{V}} A_{ij} (W^k)^\top \mathbf{x}_j^{k-1} \right)$$

$$\hat{\mathbf{z}}_i^k = \frac{\mathbf{z}_i^k - \boldsymbol{\mu}_B^k}{\sqrt{\boldsymbol{\nu}_B^k + \epsilon}}$$

$$\mathbf{x}_i^k = \Gamma^k \cdot \hat{\mathbf{z}}_i^k + \boldsymbol{\beta}^k,$$

where $\sigma$ denotes a non-linearity like the sigmoid, $\Gamma^k \in \mathbb{R}^{d_k \times d_k}$ is a diagonal matrix and $\boldsymbol{\beta}^k \in \mathbb{R}^{d_k}$ is a vector, and $\boldsymbol{\mu}_B^k, \boldsymbol{\nu}_B^k \in \mathbb{R}^{d_k}$ are vectors containing the dimension-wise mean and variance values over a mini-batch $B$ (division while computing $\hat{\mathbf{z}}_i^k$ is performed element-wise). The parameter $E^k$ is taken to collect parameters contained in all the above operations i.e. $E^k = \left\{ W^k, \Gamma^k, \boldsymbol{\beta}^k \right\}$ in the above example ( $\boldsymbol{\mu}_B^k, \boldsymbol{\nu}_B^k$ are computed using samples in a mini-batch itself). Downstream calculations in Definition 1 and Lemma 1 continue to hold with no changes.

**Incorporating Virtual Nodes into IGLU**: Virtual nodes can also be seamlessly incorporated simply by re-parameterizing the layer-wise parameter $E^k$. We are referring to (Pei et al., 2020) [Pei et al ICLR 2020] for this discussion. Let $R$ be the set of relations and let $\mathcal{N}_g(\cdot), \mathcal{N}_s(\cdot)$ denote graph neighborhood and latent-space neighborhood functions respectively. A concrete example is given below to illustrate how virtual nodes can be incorporated. We note that operations like BatchNorm etc can be additionally incorporated and alternate aggregation operations e.g. concatenation can be used instead.

$$\mathbf{g}_i^{(k,r)} = \sigma \left( \sum_{\substack{j \in \mathcal{N}_g(i) \\ \tau(i,j)=r}} A_{ij} (L_g^k)^\top \mathbf{x}_j^{k-1} \right) \qquad \text{(Low-level aggregation)}$$

$$\mathbf{s}_i^{(k,r)} = \sigma \left( \sum_{\substack{j \in \mathcal{N}_s(i) \\ \tau(i,j)=r}} T_{ij} (L_s^k)^\top \mathbf{x}_j^{k-1} \right)$$

$$\mathbf{m}_i^k = \frac{1}{|R|} \sum_{r \in R} \left( \mathbf{g}_i^{(k,r)} + \mathbf{s}_i^{(k,r)} \right) \qquad \text{(High-level aggregation)}$$

$$\mathbf{x}_i^k = \sigma \left( (W^k)^\top \mathbf{m}_i^k \right) \qquad \text{(Non-linear transform)}$$

where $\tau$ denotes the relationship indicator, $A_{ij}$ denotes the graph edge weight between nodes $i$ and $j$ and $T_{ij}$ denotes their geometric similarity in latent space. Note that $\mathbf{g}_i^{(k,r)}, \mathbf{s}_i^{(k,r)}$ corresponds to embeddings of the virtual nodes in the above example. To implement the above, the parameter $E^k$ can be taken to collect the learnable parameters contained in all the above operations i.e. $E^k = \left\{ L_g^k, L_s^k, W^k \right\}$ in the above example. Downstream calculations in Definition 1 and Lemma 1 continue to hold as is with no changes.

**Incorporating Skip/Residual Connections into IGLU**: The architecture style presented in Section 3 does not directly allow skip connections but they can be incorporated readily with no change to Definition 1 and minor changes to Lemma 1. Let us introduce the notation $k \to m$ to denote a direct forward (skip) connection directed from layer $k$ to some layer $m > k$. In a purely feed-forward style architecture, we would only have connections of the form $k \to k+1$. The following gives a simple example of a GCN with a connection that skips two layers, specifically $(k-2) \to k$.

$$\mathbf{x}_i^k = f \left( \mathbf{x}_j^{k-1}, j \in \{i\} \cup \mathcal{N}(i); E^k \right) + \mathbf{x}_i^{k-2},$$

where $f$ includes the aggregation operation with the graph Laplacian, the transformation weight matrix and any non-linearity. Definition 1 needs no changes to incorporate such architectures. Part 1 of Lemma 1 also needs no changes to address such cases. Part 2 needs a simple modification as shown below

**Lemma 3** (Lemma 1.2 adapted to skip connections). *For the final layer, we continue to have (i.e. no change)* $\boldsymbol{\alpha}^K = \mathbf{G}(W^{K+1})^\top$. *For any $k < K$, we have* $\boldsymbol{\alpha}^k = \sum_{m:k \to m} \left. \frac{\partial(\boldsymbol{\alpha}^m \odot X^m)}{\partial X^k} \right|_{all\ \boldsymbol{\alpha}^m\ s.t.\ k \to m}$ *i.e.* $\alpha_{jp}^k = \sum_{m:k \to m} \sum_{i \in \mathcal{V}} \sum_{q=1}^{d_m} \alpha_{iq}^m \cdot \frac{\partial X_{iq}^m}{\partial X_{jp}^k}$.

Note that as per the convention established in the paper, $\sum_{m:k \to m} \left. \frac{\partial(\boldsymbol{\alpha}^m \odot X^m)}{\partial X^k} \right|_{all\ \boldsymbol{\alpha}^m\ s.t.\ k \to m}$ implies that while taking the derivatives, $\boldsymbol{\alpha}^m$ values are fixed (treated as a constant) for all $m > k$ such that $k \to m$. This "conditioning" is important since $\boldsymbol{\alpha}^m$ also indirectly depends on $X^k$ if $m > k$.

**Proof of Lemma 1.2 adapted to skip connections**: We consider two cases yet again and use Definition 1 that tells us that

$$\alpha_{jp}^k = \sum_{i \in \mathcal{V}} \sum_{c \in [C]} g_{ic} \cdot \frac{\partial \hat{y}_{ic}}{\partial X_{jp}^K}$$

**Case 1** ($k = K$): Since this is the top-most layer and there are no connections going ahead let alone skipping ahead, the analysis of this case remains unchanged and continues to yield $\alpha^K = \mathbf{G}(W^{K+1})^\top$.

**Case 2** ($k < K$): Using Definition 1 and incorporating all layers to which layer $k$ has a direct or skip connection gives us

$$\alpha_{jp}^k = \sum_{i \in \mathcal{V}} \sum_{c \in [C]} g_{ic} \cdot \frac{\partial \hat{y}_{ic}}{\partial X_{jp}^k} = \sum_{m:k \to m} \sum_{i \in \mathcal{V}} \sum_{c \in [C]} g_{ic} \cdot \sum_{l \in \mathcal{V}} \sum_{q=1}^{d_m} \frac{\partial \hat{y}_{ic}}{\partial X_{lq}^m} \frac{\partial X_{lq}^m}{\partial X_{jp}^k}$$

Rearranging the terms gives us

$$\alpha_{jp}^k = \sum_{m:k \to m} \sum_{l \in \mathcal{V}} \sum_{q=1}^{d_m} \left( \sum_{i \in \mathcal{V}} \sum_{c \in [C]} g_{ic} \cdot \frac{\partial \hat{y}_{ic}}{\partial X_{lq}^m} \right) \cdot \frac{\partial X_{lq}^m}{\partial X_{jp}^k} = \sum_{m:k \to m} \sum_{l \in \mathcal{V}} \sum_{q=1}^{d_m} \alpha_{lq}^m \cdot \frac{\partial X_{lq}^m}{\partial X_{jp}^k},$$

where we simply used Definition 1 in the second step. However, the resulting term simply gives us $\alpha_{jp}^k = \sum_{m:k \to m} \left. \frac{\partial(\boldsymbol{\alpha}^m \odot X^m)}{\partial X_{jp}^k} \right|_{all\ \boldsymbol{\alpha}^m\ s.t.\ k \to m}$ which conditions on, or treats as a constant, the term $\boldsymbol{\alpha}^m$ for all $m > k$ such that $k \to m$ according to our notation convention. This finishes the proof of part 2 adapted to skip connections.

## A.6 Comparison of IGLU with VR-GCN, MVS-GNN and GNNAutoScale

We highlight the key difference between IGLU and earlier works that cache intermediate results for speeding up GNN training below.

### A.6.1 VRGCN V/S IGLU

1. **Update of Cached Variables**: VR-GCN (Chen et al., 2018a) caches only historical embeddings, and while processing a single mini-batch these historical embeddings are updated for a sampled subset of the nodes. In contrast IGLU does not update any intermediate results after processing each mini-batch. These are updated only once per epoch, after all parameters for individual layers have been updated.

2. **Update of Model Parameters**: VR-GCN's backpropagation step involves update of model parameters of all layers after each mini-batch. In contrast IGLU updates parameters of only a single layer at a time.

3. **Variance due to Sampling**: VR-GCN incurs additional variance due to neighborhood sampling which is then reduced by utilizing historical embeddings for some nodes and by computing exact embeddings for the others. IGLU does not incur such variance since IGLU uses all the neighbors.

### A.6.2 MVS-GNN V/S IGLU

MVS-GNN (Cong et al., 2020) is another work that caches historical embeddings. It follows a nested training strategy wherein firstly a large batch of nodes are sampled and mini-batches are further

created from this large batch for training. MVS-GNN handles variance due to this mini-batch creation by performing importance weighted sampling to construct mini-batches.

1. **Update of Cached Variables and Variance due to Sampling:** Building upon VR-GCN, to reduce the variance in embeddings due to its sampling of nodes at different layers, MVS-GNN caches only embeddings and uses historical embeddings for some nodes and recompute the embeddings for the others. Similar to VR-GCN, these historical embeddings are updated as and when they are part of the mini-batch used for training. As discussed above, IGLU does not incur such variance since IGLU uses all the neighbors.

2. **Update of Model Parameters:** Update of model parameters in MVS-GNN is similar to that of VR-GCN, where backpropagation step involves update of model parameters of all layers for each mini-batch. As described already, IGLU updates parameters of only a single layer at a time.

### A.6.3  GNNAUTOSCALE V/S IGLU

GNNAutoScale (Fey et al., 2021) extends the idea of caching historical embeddings from VR-GCN and provides a scalable solution.

1. **Update of intermediate representations and model parameters:** While processing a minibatch of nodes, GNNAutoScale computes the embeddings for these nodes at each layer while using historical embeddings for the immediate neighbors outside the current minibatch. After processing each mini-batch, GNNAutoScale updates the historical embeddings for nodes considered in the mini-batch. Similar to VR-GCN and MVS-GNN, GNNAutoScale updates all parameters at all layers while processing a mini-batch of nodes. In contrast IGLU does not update intermediate results (intermediate representations in Algorithm 1 and incomplete gradients in Algorithm 2) after processing each minibatch. In fact, these are updated only once per epoch, after all parameters for individual layers have been updated.

2. **Partitioning:** GNNAutoScale relies on the METIS clustering algorithm for creating mini-batches that minimize inter-connectivity across batches. This is done to minimize access to historical embeddings and reduce staleness. This algorithm tends to bring similar nodes together, potentially resulting in the distributions of clusters being different from the original dataset. This may lead to biased estimates of the full gradients while training using mini-batch SGD as discussed in Section 3.2, Page 5 of Cluster-GCN (Chiang et al., 2019). IGLU does not rely on such algorithms since it's parameter updates are concerned with only a single layer and also avoids potential additional bias.

**Similarity of IGLU with GNNAutoScale:** Both of the methods avoid a neighborhood sampling step, thereby avoiding additional variance due to neighborhood sampling and making use of all the edges in the graph. Both IGLU and GNNAutoScale propose methods to reduce the neighborhood explosion problem, although in fundamentally different manners. GNNAutoScale does so by pruning the computation graph by using historical embeddings for neighbors across different layers. IGLU on the other hand restricts the parameter updates to a single layer at a time by analyzing the gradient structure of GNNs therefore alleviating the neighborhood explosion problem.

### A.6.4  SUMMARY OF IGLU'S TECHNICAL NOVELTY AND CONTRAST WITH CACHING BASED RELATED WORKS

To summarize, IGLU is fundamentally different from these methods that cache historical embeddings in that it changes the entire training procedure of GCNs in contrast with the aforementioned caching based methods as follows:

- The above methods still follow standard **SGD style training of GCNs** in that they update the model parameters at all the layers after each mini-batch. This is very different from IGLU's parameter updates that concern **only a single layer** while processing a mini-batch.

- IGLU can cache either **incomplete gradients *or* embeddings** which is different from the other approaches that cache **only embeddings.** This provides alternate approaches for training GCNs and we demonstrate empirically that caching incomplete gradients, in fact, offers superior performance and convergence.

- Unlike GNNAutoScale and VR-GCN that update some of the historical embeddings after each mini-batch is processed, IGLU's **caching is much more aggressive** and the stale variables are updated **only once per epoch**, after all parameters for all layers have been updated.

- Theoretically, we provide **good convergence rates and bounded bias** even while using **stale gradients**, which has not been discussed in any prior works.

These are the key technical novelties of our proposed method and they are a consequence of a careful understanding of the gradient structure of GCN's themselves.

### A.6.5 EMPIRICAL COMPARISON WITH GNNAUTOSCALE

We provide an empirical comparison of IGLU with GNNAutoScale and summarize the results in Table 7 and Figure 6. It is important to note that the best results for GNNAutoScale as reported by the authors in the paper, correspond to varying hyperparameters such as number of GNN layers and different embedding dimensions across methods, datasets and architectures. However, for the experiments covered in the main paper, we use 2 layer settings for PPI-Large, Flickr and Reddit and 3 layer settings for OGBN-Arxiv and OGBN-Proteins datasets consistently for IGLU and the baseline methods, as motivated by literature. We also ensure that the embedding dimensions are uniform across IGLU and the baselines. Therefore, to ensure a fair comparison, we perform additional experiments with these parameters for GNNAutoScale set to values that are consistent with our experiments for IGLU and the baselines. We train GNNAutoScale with three variants, namely GCN, GCNII (Chen et al., 2020) and PNA (Corso et al., 2020) and report the results for each of the variant. We also note here that GNNAutoScale was implemented in PyTorch (Paszke et al., 2019) while IGLU was implemented in TensorFlow (Abadi et al., 2016). While this makes a wall-clock time comparison unsuitable as discussed in Appendix B.2, we still provide a wall-clock time comparison for completeness. We also include the best performance numbers for GNNAutoScale on these datasets (as reported by the authors in Table 5, Page 9 of the GNNAutoScale paper) across different architectures. Note that we do not provide comparisons on the OGBN-Proteins dataset since we ran into errors while trying to incorporate the dataset into the official implementation of GNNAutoScale.

**Results:** Figure 6 provides convergence plots comparing IGLU with the different architectures of GNNAutoScale and Table 7 summarizes the test performance on PPI-Large, Flickr, Reddit and OGBN-Arxiv (transductive) datasets. From the table, we observe that IGLU offers competitive performance compared to the GCN variant of GAS for the majority of the datasets. We also observe from Figure 6 that IGLU offers significant improvements in training time with rapid early convergence on the validation set. We note that more complex architectures such as GCNII and PNA offer improvements in performance to GNNAutoScale. IGLU being architecture agnostic can be incorporated with these architectures for further improvements in performance. We leave this as an avenue for future work.

### A.7 DETAILED DESCRIPTION OF ALGORITHMS 1 AND 2

We present Algorithms 1 and 2 again below (as Algorithms 3 and 4 respectively) with details of each step.

**IGLU: backprop order** Algorithm 3 implements the IGLU algorithm in its backprop variant. Node embeddings $X^k, k \in [K]$ are calculated and kept stale. They are not updated even when model parameters $E^k, k \in [K]$ get updated during the epoch. On the other hand, the incomplete task gradients $\boldsymbol{\alpha}^k, k \in [K]$ are kept refreshed using the recursive formulae given in Lemma 1. For sake of simplicity, the algorithm been presented with staleness duration of one epoch i.e. $X^k$ are refreshed at the beginning of each epoch. Variants employing shorter or longer duration of staleness can be also explored simply by updating $X^k, k \in [K]$ say twice in an epoch or else once every two epochs.

**IGLU: inverted order** Algorithm 4 implements the IGLU algorithm in its inverted variant. Incomplete task gradients $\boldsymbol{\alpha}^k, k \in [K]$ are calculated once at the beginning of every epoch and kept stale. They are not updated even when node embeddings $X^k, k \in [K]$ get updated during the epoch. On the other hand, the node embeddings $X^k, k \in [K]$ are kept refreshed. For sake of simplicity, the algorithm been presented with staleness duration of one epoch i.e. $\boldsymbol{\alpha}^k$ are refreshed at the beginning of each epoch. Variants employing shorter or longer durations of staleness can be also explored

Table 7: **Test Accuracy of IGLU compared to GNNAutoScale**. * - We perform experiments using GNNAutoScale in a setting identical to IGLU with 2-layer models on PPI-Large, Reddit and Flickr datasets and 3-layer models on OGBN-Arxiv dataset (transductive) and report the performance. For completeness, we also include the best results from GNNAutoScale for comparison. We were unable to perform experiments with GNNAutoScale on the Proteins dataset, and hence omit it for comparison. We observe that IGLU performs competitively with GNNAutoScale for models like GCN on most of the datasets. IGLU being architecture agnostic can be further combined with varied architectures like GCNII and PNA to obtain gains offered by these architecture.

| Algorithm | PPI-Large | Reddit | Flickr | Arxiv (Trans) |
|---|---|---|---|---|
| Our Experiments* | | | | |
| GAS-GCN | 0.983 | 0.954 | 0.533 | 0.710 |
| GAS-GCNII | 0.969 | 0.964 | 0.539 | 0.724 |
| GAS-PNA | 0.917 | 0.970 | 0.555 | 0.714 |
| Best Results: GNNAutoScale, (From Table 5, Page 9) | | | | |
| GAS-GCN | 0.989 | 0.954 | 0.540 | 0.716 |
| GAS-GCNII | 0.995 | 0.967 | 0.562 | 0.730 |
| GAS-PNA | 0.994 | 0.971 | 0.566 | 0.725 |
| IGLU | $0.987 \pm 0.004$ | $0.964 \pm 0.001$ | $0.515 \pm 0.001$ | $0.719 \pm 0.002$ |

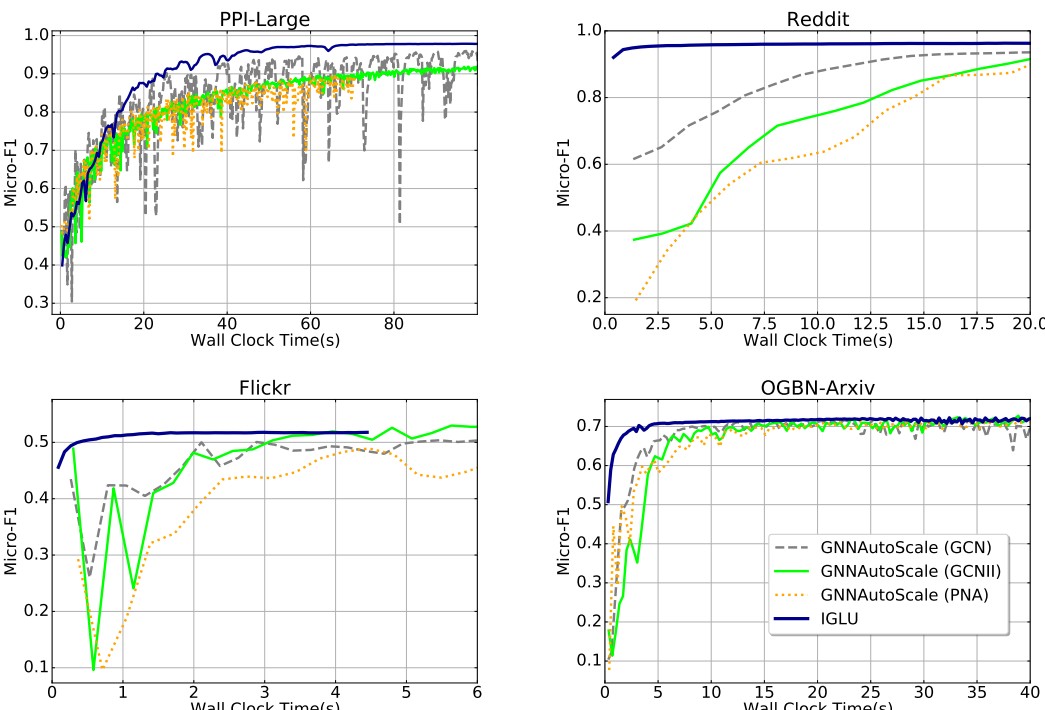

Figure 6: **Wall Clock Time vs Validation Accuracy on different datasets as compared to GN-NAutoScale.** We perform experiments using GNNAutoScale in a setting identical to IGLU with 2-layer models on PPI-Large, Reddit and Flickr datasets and 3-layer models on OGBN-Arxiv dataset and report the performance. IGLU offers competitive performance and faster convergence across the datasets.

simply by updating $\boldsymbol{\alpha}^k$ say twice in an epoch or else once every two epochs. This has been explored in Section 4.2 (see paragraph on "Analysis of Degrees of Staleness.").

---

**Algorithm 3** IGLU: backprop order

---

**Input:** GCN $\mathcal{G}$, initial features $X^0$, task loss $\mathcal{L}$
1: Initialize model parameters $E^k, k \in [K], W^{K+1}$
2: **for** epoch $= 1, 2, \ldots$ **do**
3:    **for** $k = 1 \ldots K$ **do**
4:       Refresh $X^k \leftarrow f(X^{k-1}; E^k)$   `//`$X^k$ `will be kept stale till next epoch`
5:    **end for**
6:    $\hat{\mathbf{Y}} \leftarrow X^K W^{K+1}$                                     `//Predictions`
7:    $\mathbf{G} \leftarrow \left[ \frac{\partial \ell_i}{\partial \hat{y}_{ic}} \right]_{N \times C}$               `//The loss derivative matrix`
8:    Compute $\frac{\partial \mathcal{L}}{\partial W^{K+1}} \leftarrow (X^K)^\top \mathbf{G}$       `//Using Lemma 1.1 here`
9:    Update $W^{K+1} \leftarrow W^{K+1} - \eta \cdot \frac{\partial \mathcal{L}}{\partial W^{K+1}}$
10:   Refresh $\boldsymbol{\alpha}^K \leftarrow \mathbf{G}(W^{K+1})^\top$        `//Using Lemma 1.2 here`
11:   **for** $k = K \ldots 2$ **do**
12:      Compute $\frac{\partial \mathcal{L}}{\partial E^k} \leftarrow \left. \frac{\partial (\boldsymbol{\alpha}^k \odot X^k)}{\partial E^k} \right|_{\boldsymbol{\alpha}^k}$,     `//Using Lemma 1.1 here`
13:      Update $E^k \leftarrow E^k - \eta \cdot \frac{\partial \mathcal{L}}{\partial E^k}$
14:      Refresh $\boldsymbol{\alpha}^{k-1} \leftarrow \left. \frac{\partial (\boldsymbol{\alpha}^k \odot X^k)}{\partial X^{k-1}} \right|_{\boldsymbol{\alpha}^k}$     `//Using Lemma 1.2 here`
15:   **end for**
16: **end for**

---

**Algorithm 4** IGLU: inverted order

---

**Input:** GCN $\mathcal{G}$, initial features $X^0$, task loss $\mathcal{L}$
1: Initialize model parameters $E^k, k \in [K], W^{K+1}$
2: **for** $k = 1 \ldots K$ **do**
3:    $X^k \leftarrow f(X^{k-1}; E^k)$                    `//Do an initial forward pass`
4: **end for**
5: **for** epoch $= 1, 2, \ldots$ **do**
6:    $\hat{\mathbf{Y}} \leftarrow X^K W^{K+1}$                                   `//Predictions`
7:    $\mathbf{G} \leftarrow \left[ \frac{\partial \ell_i}{\partial \hat{y}_{ic}} \right]_{N \times C}$               `//The loss derivative matrix`
     `//Use Lemma 1.2 to refresh` $\boldsymbol{\alpha}^k, k \in [K]$.
8:    Refresh $\boldsymbol{\alpha}^K \leftarrow \mathbf{G}(W^{K+1})^\top$
9:    **for** $k = K \ldots 2$ **do**
10:      Refresh $\boldsymbol{\alpha}^{k-1} \leftarrow \left. \frac{\partial (\boldsymbol{\alpha}^k \odot X^k)}{\partial X^{k-1}} \right|_{\boldsymbol{\alpha}^k}$
11:   **end for**
     `//These` $\boldsymbol{\alpha}^k, k \in [K]$ `will now be kept stale till next epoch`
12:   **for** $k = 1 \ldots K$ **do**
13:      Compute $\frac{\partial \mathcal{L}}{\partial E^k} \leftarrow \left. \frac{\partial (\boldsymbol{\alpha}^k \odot X^k)}{\partial E^k} \right|_{\boldsymbol{\alpha}^k}$,     `//Using Lemma 1.1 here`
14:      Update $E^k \leftarrow E^k - \eta \cdot \frac{\partial \mathcal{L}}{\partial E^k}$
15:      Refresh $X^k \leftarrow f(X^{k-1}; E^k)$
16:   **end for**
17:   Compute $\frac{\partial \mathcal{L}}{\partial W^{K+1}} \leftarrow (X^K)^\top \mathbf{G}$       `//Using Lemma 1.1 here`
18:   Update $W^{K+1} \leftarrow W^{K+1} - \eta \cdot \frac{\partial \mathcal{L}}{\partial W^{K+1}}$
19: **end for**

---

A.8   OGBN-PROTEINS: VALIDATION PERFORMANCE AT A MINI-BATCH LEVEL

The performance analysis in the main paper was plotted at a coarse granularity of an epoch. We refer to an epoch as one iteration of steps 6 to 18 (both inclusive) in Algorithm 4. For a finer analysis of IGLU's performance on the OGBN-Proteins dataset, we measure the Validation ROC-AUC at the granularity of a mini-batch. As mentioned in the "SGD Implementation" paragraph in Page 5 below the algorithm description in the main paper, steps 14 and 18 in Algorithm 4 are implemented using mini-batch SGD. Recall that OGBN-Proteins used a 3 layer GCN. For the proteins dataset we have $\sim$ 170 mini-batches for training. We update the parameters for each layer using all the mini-batches

from the layer closest to the input to the layer closest to the output as detailed in Algorithm 4. To generate predictions, we compute partial forward passes after the parameters of each layer is updated. We plot the validation ROC-AUC as the first epoch progresses in Figure 7. We observe that when the layer closest to the input is trained (Layer 1 in the figure), IGLU has an ROC-AUC close to 0.5 on the validation set. Subsequently once the second layer (Layer 2 in the figure) is trained, we observe that the validation ROC-AUC improves from around $\sim 0.51$ to $\sim 0.57$ and finally once the layer closest to the output is trained (Layer 3 in the figure), the ROC-AUC progresses quickly to a high validation ROC-AUC of $\sim 0.81$. In the figure in the main paper the high validation score reflects the result at the end of the first epoch. Training the GCN using a total of $\sim 510$ minibatches ($\sim 170$ per layer) approximately takes 5 seconds as depicted in Figure 2 in the main paper.

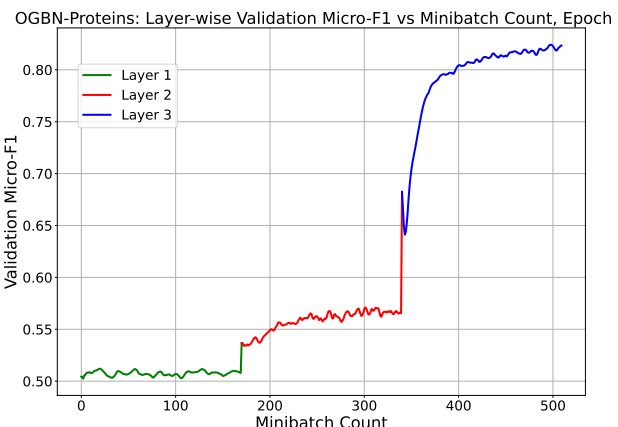

Figure 7: **Fine-grained Validation ROC-AUC for IGLU on the Proteins Dataset for Epoch 1.** We depict the value of Validation ROC - AUC at the granularity of a minibatch for the first epoch. We observe that the Validation ROC-AUC begins with a value close to 0.5 and quickly reaches an ROC-AUC of 0.81 by the end of the first epoch. As mentioned in the text, Proteins uses a 3 layer GCN and each layer processes $\sim 170$ mini-batches.

## B  DATASET STATISTICS AND ADDITIONAL EXPERIMENTAL RESULTS

### B.1  DATASET STATISTICS

Table 8 provides details on the benchmark node classification datasets used in the experiments. The following five benchmark datasets were used to empirically demonstrate the effectiveness of IGLU: predicting the communities to which different posts belong in Reddit[‡] (Hamilton et al., 2017), classifying protein functions across various biological protein-protein interaction graphs in PPI-Large[§] (Hamilton et al., 2017), categorizing types of images based on descriptions and common properties in Flickr[¶] (Zeng et al., 2020), predicting paper-paper associations in OGBN-Arxiv[‖] (Hu et al., 2020) and categorizing meaningful associations between proteins in OGBN-Proteins[**] (Hu et al., 2020).

### B.2  COMPARISON WITH ADDITIONAL BASELINES

In addition to the baselines mentioned in Table 1, Table 9 compares IGLU to LADIES (Zou et al., 2019), L2-GCN (You et al., 2020), AS-GCN (Huang et al., 2018), MVS-GNN (Cong et al., 2020), FastGCN (Chen et al., 2018b), SIGN (Frasca et al., 2020), PPRGo (Bojchevski et al., 2020) and Bandit Sampler's (Liu et al., 2020) performance on the test set. However, a wall-clock time comparison with

---

[‡]http://snap.stanford.edu/graphsage/reddit.zip

[§]http://snap.stanford.edu/graphsage/ppi.zip

[¶]https://github.com/GraphSAINT/GraphSAINT - Google Drive Link

[‖]https://ogb.stanford.edu/docs/nodeprop/#ogbn-arxiv

[**]https://ogb.stanford.edu/docs/nodeprop/#ogbn-proteins

Table 8: **Datasets used in experiments along with their statistics.** MC refers to a multi-class problem, whereas ML refers to a multi-label problem.

| Dataset | # Nodes | # Edges | Avg. Degree | # Features | # Classes | Train/Val/Test |
|---|---|---|---|---|---|---|
| PPI-Large | 56944 | 818716 | 14 | 50 | 121 (ML) | 0.79/0.11/0/10 |
| Reddit | 232965 | 11606919 | 60 | 602 | 41 (MC) | 0.66/0.10/0.24 |
| Flickr | 89250 | 899756 | 10 | 500 | 7 (MC) | 0.5/0.25/0.25 |
| OGBN-Proteins | 132534 | 39561252 | 597 | 8 | 112 (ML) | 0.65/0.16/0.19 |
| OGBN-Arxiv | 169343 | 1166243 | 13 | 128 | 40 (MC) | 0.54/0.18/0.28 |

these methods is not provided since the author implementations of LADIES, L2GCN, MVS-GNN and SIGN are in PyTorch (Paszke et al., 2019) which has been shown to be less efficient than Tensorflow (Chiang et al., 2019; Abadi et al., 2016) for GCN applications. Also, the official AS-GCN, FastGCN and Bandit Sampler implementations released by the authors were for 2 layer models only, whereas some datasets such as Proteins and Arxiv require 3 layer models for experimentation. Attempts to generalize the code to a 3 layer model ran into runtime errors, hence the missing results are denoted by ** in Table 9 and these methods are not considered for a timing analysis. MVS-GNN also runs into a runtime error on the Proteins dataset denoted by ‖. *IGLU continues to significantly outperform all additional baselines on all the datasets.*

Table 9: **Performance on Test Set for IGLU compared to additional algorithms.** The metric is ROC-AUC for Proteins and Micro-F1 for the others IGLU still retains the state-of-the-art results across all datasets even when compared to these new baselines. MVS-GNN ran into runtime error on the Proteins dataset (**denoted by ‖**). AS-GCN, FastGCN and BanditSampler run into a runtime error on datasets that require more than two layers (**denoted by ∗∗**). Please refer to section B.2 for details.

| Algorithm | PPI-Large | Reddit | Flickr | Proteins | Arxiv |
|---|---|---|---|---|---|
| LADIES | $0.548 \pm 0.011$ | $0.923 \pm 0.008$ | $0.488 \pm 0.012$ | $0.636 \pm 0.011$ | $0.667 \pm 0.002$ |
| L2GCN | $0.923 \pm 0.008$ | $0.938 \pm 0.001$ | $0.485 \pm 0.001$ | $0.531 \pm 0.001$ | $0.656 \pm 0.004$ |
| ASGCN | $0.687 \pm 0.001$ | $0.958 \pm 0.001$ | $0.504 \pm 0.002$ | ** | ** |
| MVS-GNN | $0.880 \pm 0.001$ | $0.950 \pm 0.001$ | $0.507 \pm 0.002$ | ‖ | $0.695 \pm 0.003$ |
| FastGCN | $0.513 \pm 0.032$ | $0.924 \pm 0.001$ | $0.504 \pm 0.001$ | ** | ** |
| SIGN | $0.970 \pm 0.003$ | $\mathbf{0.966 \pm 0.003}$ | $0.510 \pm 0.001$ | $0.665 \pm 0.008$ | $0.649 \pm 0.003$ |
| PPRGo | $0.626 \pm 0.002$ | $0.946 \pm 0.001$ | $0.501 \pm 0.001$ | $0.659 \pm 0.006$ | $0.678 \pm 0.003$ |
| BanditSampler | $0.905 \pm 0.003$ | $0.957 \pm 0.000$ | $0.513 \pm 0.001$ | ** | ** |
| IGLU | $\mathbf{0.987 \pm 0.004}$ | $0.964 \pm 0.001$ | $\mathbf{0.515 \pm 0.001}$ | $\mathbf{0.784 \pm 0.004}$ | $\mathbf{0.718 \pm 0.001}$ |

Table 10: **Per epoch time (in seconds) for different methods as the number of layers increase on the OGBN-Proteins dataset.** ClusterGCN ran into a runtime error on this dataset as noted earlier. VRGCN ran into a runtime error for a 4 layer model (**denoted by ‖**). IGLU and GraphSAINT scale almost linearly with the number of layers. It should be noted that these times strictly include only optimization time. GraphSAINT has a much lower per-epoch time than IGLU because of the large sizes of subgraphs per batch ($\sim 10000$ nodes), while IGLU uses minibatches of size of 512. This results in far less gradient updates within an epoch for GraphSAINT when compared with IGLU, resulting in a much smaller per-epoch time but requiring more epochs overall. Please refer to section B.3.1 for details.

| | Number of Layers | | |
|---|---|---|---|
| **Method** | **2** | **3** | **4** |
| GraphSAGE | 2.6 | 14.5 | 163.1 |
| VR-GCN | 2.3 | 21.5 | ‖ |
| GraphSAINT | 0.45 | 0.61 | 0.76 |
| IGLU | 2.97 | 5.27 | 6.99 |

Table 11: **Test Performance (ROC-AUC) at Best Validation for different methods as the number of layers increase on the OGBN-Proteins Dataset.** Results are reported for a single run but trends were observed to remain consistent across repeated runs. VRGCN ran into runtime error for 4 layers (**denoted by ‖**). IGLU offers steady increase in performance as the number of layers increase, as well as state-of-the-art performance throughout. GraphSAGE shows a decrease in performance on moving from 3 to 4 layers while GraphSAINT shows only a marginal increase in performance. Please refer to section B.3.2 for details.

|  | Number of Layers | | |
|---|---|---|---|
| **Method** | **2** | **3** | **4** |
| GraphSAGE | 0.755 | 0.759 | 0.742 |
| VR-GCN | 0.732 | 0.749 | ‖ |
| GraphSAINT | 0.752 | 0.764 | 0.767 |
| **IGLU** | **0.768** | **0.783** | **0.794** |

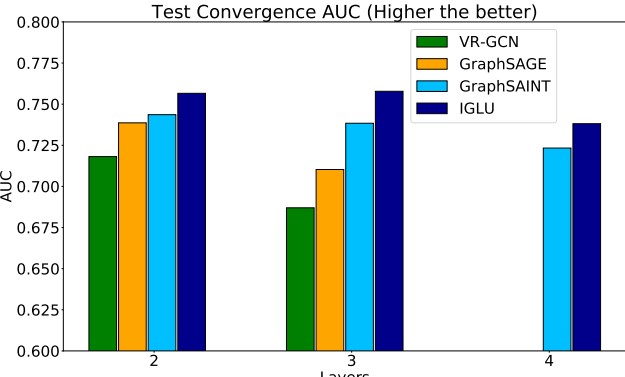

Figure 8: **Test Convergence AUC plots across different number of layers on the OGBN-Proteins dataset.** IGLU has consistently higher AUTC values compared to the other baselines, demonstrating increased stability, faster convergence and better generalization. GraphSAGE suffers from neighborhood explosion problem and the training became very slow as noted earlier. This results in a decrease in the AUTC while going from 3 to 4 layers. GraphSAGE's AUTC for 4 layers is only 0.313, and is thus not visible in the plot. VRGCN also suffers from the neighborhood explosion problem and runs into runtime errors for a 4 layer model. ClusterGCN runs into runtime error for the OGBN-Proteins for all of 2, 3 and 4 layers and is therefore not present in this analysis. Please refer to Section B.3.2 for details.

## B.3    TIMING ANALYSIS FOR SCALING TO MORE LAYERS

To compare the scalability and performance of different algorithms for deeper models, models with 2, 3 and 4 layers were trained for IGLU and the baseline methods. IGLU was observed to offer a per-epoch time that scaled roughly **linearly with the number of layers** as well as offer the **highest gain in test performance as the number of layers was increased**.

### B.3.1    PER EPOCH TRAINING TIME

Unlike neighbor sampling methods like VRGCN and GraphSAGE, IGLU does not suffer from the neighborhood explosion problem as the number of layers increases, since IGLU updates involve only a single layer at any given time. We note that GraphSAINT and ClusterGCN also do not suffer from the neighborhood explosion problem directly since they both operate by creating GCNs on subgraphs. However, these methods may be compelled to select large subgraphs or else suffer from poor convergence. To demonstrate IGLU's effectiveness in solving the neighborhood explosion problem, the per-epoch training times are summarized as a function of the number of layers in Table

10 on the Proteins dataset. A comparison with ClusterGCN could not be provided as since it ran into runtime errors on this dataset. In Table 10, while going from 2 to 4 layers, GraphSAINT was observed to require $\sim 1.6\times$ more time per epoch while IGLU required $\sim 2.3\times$ more time per epoch, with both methods scaling almost linearly with respect to number of layers as expected. However, GraphSAGE suffered a $\sim 62\times$ increase in time taken per epoch in this case, suffering from the neighborhood explosion problem. VRGCN ran into a run-time error for the 4 layer setting (denoted by || in Table 10). Nevertheless, even while going from 2 layers to 3 layers, VRGCN and GraphSAGE are clearly seen to suffer from the neighborhood explosion problem, resulting in an increase in training time per epoch of almost $\sim 9.4\times$ and $\sim 5.6\times$ respectively.

We note that the times for GraphSAINT in Table 10 are significantly smaller than those of IGLU even though earlier discussion reported IGLU as having the fastest convergence. However, there is no contradiction – GraphSAINT operates with very large subgraphs, with each subgraph having almost 10 % of the nodes of the entire training graph ($\sim 10000$ nodes in a minibatch), while IGLU operates with minibatches of size 512, resulting in IGLU performing a lot more gradient updates within an epoch, as compared with GraphSAINT. Consequently, **IGLU also takes fewer epochs to converge to a better solution than GraphSAINT**, thus compensating for the differences in time taken per epoch.

### B.3.2 TEST CONVERGENCE AUC

This section explores the effect of increasing the number of layers on convergence rate, optimization time, and final accuracy, when scaling to larger number of layers. To jointly estimate the efficiency of a method in terms of wall clock time and test performance achieved, the area under the test convergence plots (AUTC) for various methods was computed. A method that converged rapidly and that too to better performance levels would have a higher AUTC than a method that converges to suboptimal values or else converges very slowly. To fairly time all methods, each method was offered time that was triple of the time it took the best method to reach its highest validation score. Defining the cut-off time this way ensures that methods that may not have rapid early convergence still get a fair chance to improve by having better final performance, while also simultaneously penalizing methods that may have rapid early convergence but poor final performance. We rescale the wall clock time to be between 0 and 1, where 0 refers to the start of training while 1 refers to the cut-off time.

**Results:** Figure 8 summarizes the AUTC values. IGLU consistently obtains higher AUC values than all methods for all number of layers demonstrating its stability, rapid convergence during early phases of training and ability to generalize better as compared to other baselines. GraphSAGE suffered from neighborhood explosion leading to increased training times and hence decreased AUC values as the number of layers increase. VR-GCN also suffered from the same issue, and additionally ran into a run-time error for 4 layer models.

**Test Performance with Increasing Layers:** Table 11 summarizes the final test performances for different methods, across different number of layers. The performance for some methods is inconsistent as the depth increases whereas *IGLU consistently outperforms all the baselines in this case as well* with gains in performance as we increase the number of layers hence *making it an attractive technique to train deeper GCN models*.

### B.4 SCALABILITY TO LARGER DATASETS: OGBN - PRODUCTS

To demonstrate the ability of IGLU to scale to very large datasets, we performed experiments on the OGBN-Products dataset (Hu et al., 2020), one of the largest datasets in the OGB collection, with its statistics summarized in Table 12.

Table 12: **Statistics for the OGB-Products datasets**. MC refers to a multi-class problem, whereas ML refers to a multi-label problem.

| Dataset | # Nodes | # Edges | Avg. Degree | # Features | # Classes | Train/Val/Test |
|---------|---------|---------|-------------|------------|-----------|----------------|
| OGBN-Products | 2,449,029 | 61,859,140 | 50.5 | 100 | 47 (MC) | 0.08/0.02/0.90 |

To demonstrate scalability, we conducted experiments in the transductive setup since this setup involves using the full graph. In addition, this was the original setup in which the dataset was

Table 13: **Performance on the OGBN-Products Test Set for IGLU compared to baseline algorithms.** IGLU outperforms all the baseline methods on this significantly large dataset as well.

| Algorithm | Test Micro-F1 |
|---|---|
| GCN | $0.760 \pm 0.002$ |
| GraphSAGE | $0.787 \pm 0.004$ |
| ClusterGCN | $0.790 \pm 0.003$ |
| GraphSAINT | $0.791 \pm 0.002$ |
| SIGN (3,3,0) | $0.771 \pm 0.001$ |
| SIGN (5,3,0) | $0.776 \pm 0.001$ |
| IGLU | $\mathbf{0.793} \pm 0.003$ |

benchmarked, therefore allowing for a direct comparison with the baselines (results taken from Table 4 in OGB (Hu et al., 2020) and Table 6 in SIGN (Frasca et al., 2020)). We summarize the performance results in the Table 13, reporting Micro-F1 as the metric. We however do not provide timing comparisons with the baseline methods since IGLU is implemented in TensorFlow while the baselines in the original benchmark Hu et al. (2020) are implemented in PyTorch. This therefore renders a direct comparison of wall-clock time unsuitable. Please refer to Appendix B.2 for more details.

We observe that IGLU is able to scale to the OGBN-Products dataset with over 2.4 million nodes and outperforms all of the baseline methods.

## B.5 APPLICABILITY OF IGLU IN THE TRANSDUCTIVE SETTING: OGBN-ARXIV AND OGBN-PROTEINS

In addition to the results in the inductive setting reported in the main paper, we perform additional experiments on the OGBN-Arxiv and OGBN-Proteins dataset in the transductive setting to demonstrate IGLU's applicability across inductive and transductive tasks and compare the performance of IGLU to that of transductive baseline methods in Table 14 (results taken from OGB (Hu et al., 2020)).

Table 14: **Comparison of IGLU's Test performance with the baseline methods in the transductive setting on the OGBN-Arxiv and OGBN-Proteins datasets.** The metric is Micro-F1 for OGBN-Arxiv and ROC-AUC for OGBN-Proteins.

| Algorithm | OGBN-Arxiv | OGBN-Proteins |
|---|---|---|
| GCN | $0.7174 \pm 0.0029$ | $0.7251 \pm 0.0035$ |
| GraphSAGE | $0.7149 \pm 0.0027$ | $0.7768 \pm 0.0020$ |
| IGLU | $\mathbf{0.7193} \pm 0.0018$ | $\mathbf{0.7840} \pm 0.0061$ |

We observe that even in the transductive setting, IGLU outperforms the baseline methods on both the OGBN-Arxiv and OGBN-Proteins datasets.

## B.6 ARCHITECTURE AGNOSTIC NATURE OF IGLU

To demonstrate the applicability of IGLU to a wide-variety of architectures, we perform experiments on IGLU with Graph Attention Networks (GAT) (Veličković et al., 2018), GCN (Kipf & Welling, 2017) and GraphSAGE (Hamilton et al., 2017) based architectures and summarize the results for the same in Table 15 and 16 respectively as compared to these baseline methods. We use the Cora, Citeseer and Pubmed datasets as originally used in Veličković et al. (2018) for comparison with the GAT based architecture and the OGBN-Arxiv dataset for comparison with GCN and GraphSAGE based architectures (baseline results as originally reported in (Hu et al., 2020)).

We observe that the IGLU+GAT, IGLU+GCN and IGLU+GraphSAGE outperforms the baseline methods across datasets, thereby demonstrating the architecture agnostic nature of IGLU.

Table 15: **Comparison of IGLU +GAT's test performance with the baseline GAT on different datasets.**

| Algorithm | Cora | Citeseer | Pubmed |
|---|---|---|---|
| GAT | $0.823 \pm 0.007$ | $0.711 \pm 0.006$ | $0.786 \pm 0.004$ |
| **IGLU + GAT** | **0.829** $\pm 0.004$ | **0.717** $\pm 0.005$ | **0.787** $\pm 0.002$ |

Table 16: **Comparison of IGLU's test performance with GCN and GraphSAGE architectures with the baseline methods on the OGBN-Arxiv dataset.**

| Algorithm | OGBN-Arxiv |
|---|---|
| GCN | $0.7174 \pm 0.0029$ |
| GraphSAGE | $0.7149 \pm 0.0027$ |
| **IGLU + GCN** | **0.7187** $\pm 0.0014$ |
| **IGLU + GraphSAGE** | **0.7155** $\pm 0.0032$ |

### B.7 EXPERIMENTS USING A SMOOTH ACTIVATION FUNCTION: GELU

To understand the effect of using non-smooth vs smooth activation functions on IGLU, we perform experiments using the GELU (Hendrycks & Gimpel, 2020) activation function which is a smooth function and in-line with the objective smoothness assumptions made in Theorem 2.

$$\text{GELU}(x) = xP(X \leq x) = x\Phi(x) = x \cdot \frac{1}{2}[1 + \text{erf}(x/\sqrt{2})]$$

We compare the performance of IGLU using ReLU and GELU on all the 5 datasets in the main paper and summarize the results in Table 17.

Table 17: **Effect of Non-smooth vs Smooth activation functions: Test Performance of IGLU on different datasets using ReLU and GELU activation functions.** Metrics are the same for the datasets as reported in Table 1 of the main paper. Results reported are averaged over five different runs.

| Dataset | ReLU | GELU |
|---|---|---|
| PPI-Large | $0.987 \pm 0.004$ | $0.987 \pm 0.000$ |
| Reddit | $0.964 \pm 0.001$ | $0.962 \pm 0.000$ |
| Flickr | $0.515 \pm 0.001$ | $0.516 \pm 0.001$ |
| Proteins | $0.784 \pm 0.004$ | $0.782 \pm 0.002$ |
| Arxiv | $0.718 \pm 0.001$ | $0.720 \pm 0.002$ |

We observe that IGLU is able to enjoy very similar performance across both GELU and ReLU as the activation functions, thereby justifying the practicality of our smoothness assumptions.

### B.8 ANALYSIS ON DEGREE OF STALENESS: BACKPROP ORDER OF UPDATES

To understand the effect of more frequent updates in the backprop variant, we performed additional experiments using the backprop variant on the PPI-Large dataset and varied the frequency of updates, the results of which are reported in Table 18 for a single run. We fix the hyperparameters and train for 200 epochs.

Table 18: **Accuracy vs different update frequency on PPI-Large: Backprop Order**

| Update Frequency | Train Micro-F1 | Validation Micro-F1 | Test Micro-F1 |
|---|---|---|---|
| 0.5 | 0.761 | 0.739 | 0.756 |
| 1 | 0.805 | 0.778 | 0.796 |
| 2 | 0.794 | 0.769 | 0.784 |

We observe that more frequent updates help stabilize training better. We also observe that update frequencies 1 and 2 perform competitively, and both significantly outperform update frequency 0.5.

However we note that with higher update frequency, we incur an additional computational cost since we need to re-compute embeddings more frequently. We believe that both improved stability and competitive performance can be attributed to the fresher embeddings, which are otherwise kept stale

within an epoch in this order of updates. The experiment with frequency 0.5 has a slower convergence and comparatively poor performance as expected.

### B.9 CONVERGENCE - TRAIN LOSS VS WALL CLOCK TIME

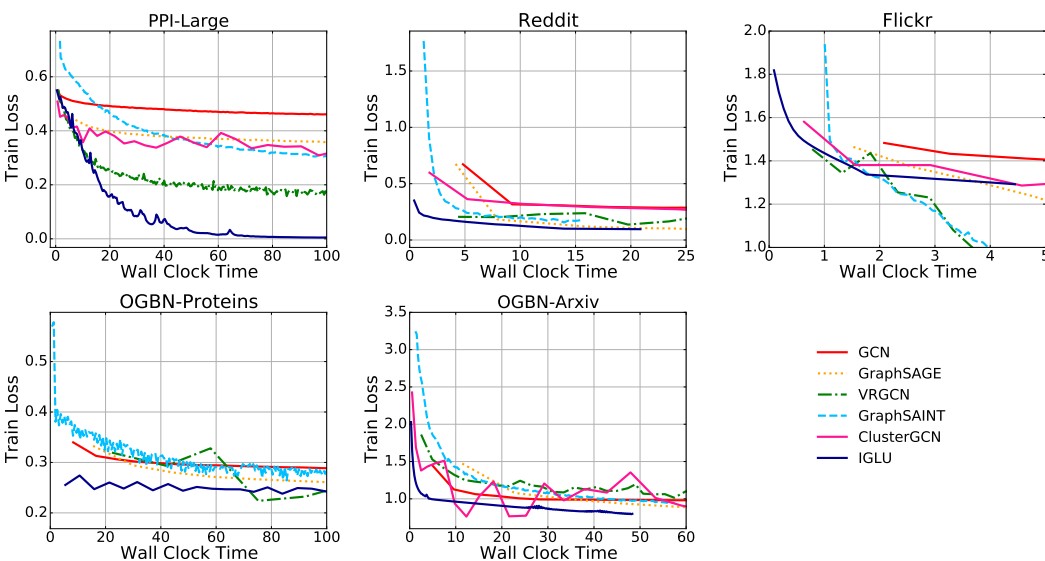

Figure 9: **Training Loss curves of different methods on the benchmark datasets against Wall clock time.**

Figure 9 provides train loss curves for all datasets and methods in Table 1. IGLU is able to achieve a lower training loss faster compared to all the baselines across datasets.

## C    THEORETICAL PROOFS

In this section we provide a formal restatement of Theorem 2 as well as its proof. We also provide the proof of Lemma 1 below.

### C.1    PROOF FOR LEMMA 1

**Proof of Lemma 1** We consider the two parts of Lemma 1 separately and consider two cases while proving each part. For each part, Case 1 pertains to the final layer and Case 2 considers intermediate layers.

**Clarification about some Notation in the statements of Definition 1 and Lemma 1**: The notation $\frac{\partial(\mathbf{G} \odot \hat{\mathbf{Y}})}{\partial X_{jp}^k}\Big|_{\mathbf{G}}$ is meant to denote the partial derivative w.r.t $X_{jp}^k$ but while keeping $\mathbf{G}$ fixed i.e. treated as a constant or being "conditioned upon" (indeed both $\mathbf{G}$ and $\hat{\mathbf{Y}}$ depend on $X_{jp}^k$ but the definition keeps $\mathbf{G}$ fixed while taking derivatives). Similarly, in Lemma 1 part 1, $\boldsymbol{\alpha}^k$ is fixed (treated as a constant) in the derivative in the definition of $\partial \mathcal{L}/\partial E^k$ and in part 2, $\boldsymbol{\alpha}^{k+1}$ is fixed (treated as a constant) in the derivative in the definition of $\boldsymbol{\alpha}^k$.

**Recalling some Notation for sake of completeness**: We recall that $\mathbf{x}_i^k$ is the $k$-th layer embedding of node $i$. $\mathbf{y}_i$ is the $C$-dimensional ground-truth label vector for node $i$ and $\hat{\mathbf{y}}_i$ is the $C$-dimensional predicted score vector for node $i$. $K$ is total number of layers in the GCN. $\mathcal{L}_i$ denotes the loss on the $i$-th node and $\mathcal{L}$ denotes the total loss summed over all nodes. $d_k$ is the embedding dimensionality after the $k$-th layer.

**Proof of Lemma 1.1** We analyze two cases

**Case 1** ($k = K$ i.e. final layer): Recall that the predictions for node $i$ are obtained as $\hat{\mathbf{y}}_i = (W^{K+1})^\top x_i^K$. Thus we have

$$\frac{\partial \mathcal{L}}{\partial W^{K+1}} = \sum_{i \in \mathcal{V}} \sum_{c \in [C]} \frac{\partial \mathcal{L}_i}{\partial \hat{y}_{ic}} \frac{\partial \hat{y}_{ic}}{\partial W^{K+1}}.$$

Now, $\frac{\partial \mathcal{L}_i}{\partial \hat{y}_{ic}} = g_{ic}$ by definition. If we let $\mathbf{w}_c^{K+1}$ denote the $c$-th column of the $d_k \times C$ matrix $W^{K+1}$, then it is clear that $\hat{y}_{ic}$ depends only on $\mathbf{w}_c^{K+1}$ and $\mathbf{x}_i^K$. Thus, we have

$$\frac{\partial \hat{y}_{ic}}{\partial W^{K+1}} = \frac{\partial \hat{y}_{ic}}{\partial \mathbf{w}_c^{K+1}} \cdot \mathbf{e}_c^\top = (\mathbf{x}_i^K)^\top \mathbf{e}_c^\top,$$

where $\mathbf{e}_c$ is the $c$-th canonical vector in $C$-dimensions with 1 at the $c$-th coordinate and 0 everywhere else. This gives us

$$\frac{\partial \mathcal{L}}{\partial W^{K+1}} = \sum_{i \in \mathcal{V}} (\mathbf{x}_i^K)^\top \sum_{c \in [C]} g_{ic} \cdot \mathbf{e}_c^\top = (X^K)^\top \mathbf{G}$$

**Case 2** ($k < K$ i.e. intermediate layers): We recall that $X^k$ stacks all $k$-th layer embeddings as an $N \times d_k$ matrix and $X^k = f(X^{k-1}; E^k)$ where $E^k$ denotes the parameters (weights, offsets, scales) of the k-th layer. Thus we have

$$\frac{\partial \mathcal{L}}{\partial E^k} = \sum_{i \in \mathcal{V}} \sum_{c \in [C]} \frac{\partial \mathcal{L}}{\partial \hat{y}_{ic}} \frac{\partial \hat{y}_{ic}}{\partial E^k}.$$

As before, $\frac{\partial \mathcal{L}}{\partial \hat{y}_{ic}} = g_{ic}$ by definition and we have

$$\frac{\partial \hat{y}_{ic}}{\partial E^k} = \sum_{j \in \mathcal{V}} \sum_{p=1}^{d_k} \frac{\partial \hat{y}_{ic}}{\partial X_{jp}^k} \frac{\partial X_{jp}^k}{\partial E^k}$$

This gives us

$$\frac{\partial \mathcal{L}}{\partial E^k} = \sum_{i \in \mathcal{V}} \sum_{c \in [C]} g_{ic} \cdot \sum_{j \in \mathcal{V}} \sum_{p=1}^{d_k} \frac{\partial \hat{y}_{ic}}{\partial X_{jp}^k} \frac{\partial X_{jp}^k}{\partial E^k} = \sum_{j \in \mathcal{V}} \sum_{p=1}^{d_k} \alpha_{jp}^k \cdot \frac{\partial X_{jp}^k}{\partial E^k} = \left. \frac{\partial (\boldsymbol{\alpha}^k \odot X^k)}{\partial E^k} \right|_{\boldsymbol{\alpha}^k},$$

where we used the definition of $\alpha_{jp}^k$ in the second step and used a "conditional" notation to get the third step. We reiterate that $\left. \frac{\partial (\boldsymbol{\alpha}^k \odot X^k)}{\partial E^k} \right|_{\boldsymbol{\alpha}^k}$ implies that $\boldsymbol{\alpha}^k$ is fixed (treated as a constant) while taking the derivative. This "conditioning" is critical since $\boldsymbol{\alpha}^k$ also depends on $E^k$. This concludes the proof.

**Proof of Lemma 1.2**: We consider two cases yet again and use Definition 1 that tells us that

$$\alpha_{jp}^k = \sum_{i \in \mathcal{V}} \sum_{c \in [C]} g_{ic} \cdot \frac{\partial \hat{y}_{ic}}{\partial X_{jp}^k}$$

**Case 1** ($k = K$): Since $\hat{\mathbf{y}}_i = (W^{K+1})^\top \mathbf{x}_i^K$, we know that $\hat{y}_{ic}$ depends only on $\mathbf{x}_i^K$ and $\mathbf{w}_c^{K+1}$ where as before, $\mathbf{w}_c^{K+1}$ is the $c$-th column of the matrix $W^{K+1}$. This gives us $\frac{\partial \hat{y}_{ic}}{\partial X_{jp}^K} = 0$ if $i \neq j$ and $\frac{\partial \hat{y}_{ic}}{\partial X_{jp}^K} = w_{pc}^K$ if $i = j$ where $w_{pc}^K$ is the $(p, c)$-th entry of the matrix $W^{K+1}$ (or in other words, the $p$-th coordinate of the vector $\mathbf{w}_c^{K+1}$). This tells us that

$$\alpha_{jp}^K = \sum_{i \in \mathcal{V}} \sum_{c \in [C]} g_{ic} \cdot \frac{\partial \hat{y}_{ic}}{\partial X_{jp}^K} = \sum_{c \in [C]} g_{jc} \cdot w_{pc}^K,$$

which gives us $\boldsymbol{\alpha}^K = \mathbf{G}(W^{K+1})^\top$.

**Case 2** ($k < K$): By Definition 1 we have

$$\alpha_{jp}^k = \sum_{i \in \mathcal{V}} \sum_{c \in [C]} g_{ic} \cdot \frac{\partial \hat{y}_{ic}}{\partial X_{jp}^k} = \sum_{i \in \mathcal{V}} \sum_{c \in [C]} g_{ic} \cdot \sum_{l \in \mathcal{V}} \sum_{q=1}^{d_{k+1}} \frac{\partial \hat{y}_{ic}}{\partial X_{lq}^{k+1}} \frac{\partial X_{lq}^{k+1}}{\partial X_{jp}^k}$$

Rearranging the terms gives us

$$\alpha_{jp}^k = \sum_{l \in \mathcal{V}} \sum_{q=1}^{d_{k+1}} \left( \sum_{i \in \mathcal{V}} \sum_{c \in [C]} g_{ic} \cdot \frac{\partial \hat{y}_{ic}}{\partial X_{lq}^{k+1}} \right) \cdot \frac{\partial X_{lq}^{k+1}}{\partial X_{jp}^k} = \sum_{l \in \mathcal{V}} \sum_{q=1}^{d_{k+1}} \alpha_{lq}^{k+1} \cdot \frac{\partial X_{lq}^{k+1}}{\partial X_{jp}^k},$$

where we simply used Definition 1 in the second step. However, the resulting term is simply $\left. \frac{\partial (\boldsymbol{\alpha}^{k+1} \odot X^{k+1})}{\partial X_{jp}^k} \right|_{\boldsymbol{\alpha}^{k+1}}$ which conditions on, or treats as a constant, the term $\boldsymbol{\alpha}^{k+1}$ according to our notation convention. This finishes the proof of part 2.

## C.2 Statement of Convergence Guarantee

The rate for full-batch updates, as derived below, is $\mathcal{O}\left(\frac{1}{T^{\frac{2}{3}}}\right)$. This *fast* rate offered by full-batch updates is asymptotically superior to the $\mathcal{O}\left(\frac{1}{\sqrt{T}}\right)$ rate offered by mini-batch SGD updates. This is due to the additional variance due to mini-batch construction that mini-batch SGD variants have to incur.

**Theorem 4** (IGLU Convergence (Final)). *Suppose the task loss function $\mathcal{L}$ has $H$-smooth and an architecture that offers bounded gradients and Lipschitz gradients as quantified below, then if IGLU in its inverted variant (Algorithm 2) is executed with step length $\eta$ and a staleness count of $\tau$ updates per layer as in steps 7, 10 in Algorithm 2, then within $T$ iterations, we must have*

1. $\|\nabla \mathcal{L}\|_2^2 \leq \mathcal{O}\left(1/T^{\frac{2}{3}}\right)$ *if model update steps are carried out on the entire graph in a full-batch with step length $\eta = \mathcal{O}\left(1/T^{\frac{1}{3}}\right)$ and $\tau = \mathcal{O}(1)$.*

2. $\|\nabla \mathcal{L}\|_2^2 \leq \mathcal{O}\left(1/\sqrt{T}\right)$ *if model update steps are carried out using mini-batch SGD with step length $\eta = \mathcal{O}\left(1/\sqrt{T}\right)$ and $\tau = \mathcal{O}\left(T^{\frac{1}{4}}\right)$.*

It is curious that the above result predicts that when using mini-batch SGD, a non-trivial amount of staleness ($\tau = \mathcal{O}\left(T^{\frac{1}{4}}\right)$ as per the above result) may be optimal and premature refreshing of embeddings/incomplete gradients may be suboptimal as was also seen in experiments reported in Table 2 and Figure 4. Our overall proof strategy is the following

1. **Step 1**: Analyze how lazy updates in IGLU affect model gradients
2. **Step 2**: Bound the bias in model gradients in terms of staleness due to the lazy updates
3. **Step 3**: Using various properties such as smoothness and boundedness of gradients, obtain an upper bound for the bias in the gradients in terms of number of iterations since last update
4. **Step 4**: Use the above to establish the convergence guarantee

We will show the results for the variant of IGLU that uses the *inverted* order of updates as given in Algorithm 2. A similar proof technique will also work for the variant that uses the *backprop* order of updates. Also, to avoid clutter, we will from hereon assume that a normalized total loss function is used for training i.e. $\mathcal{L} = \frac{1}{N} \sum_{i \in \mathcal{V}} \ell_i$ where $N = |\mathcal{V}|$ is the number of nodes in the training graph.

## C.3 Step 1: Partial Staleness and its Effect on Model Gradients

A peculiarity of the inverted order of updates is that the embeddings $X^k, k \in [K]$ are never stale in this variant. To see this, we use a simple inductive argument. The base case of $X^0$ is obvious – it is never stale since it is never meant to be updated. For the inductive case, notice how, the moment any parameter $E^k$ is updated in step 7 of Algorithm 2 (whether by mini-batch SGD or by full-batch GD), immediately thereafter in step 8 of the algorithm, $X^k$ is updated using the current value of $X^{k-1}$ and $E^k$. Since by induction $X^{k-1}$ never has a stale value, this shows that $X^k$ is never stale either, completing the inductive argument.

This has an interesting consequence: by Lemma 1, we have $\frac{\partial \mathcal{L}}{\partial E^k} = \frac{1}{N} \left. \frac{\partial(\boldsymbol{\alpha}^k \odot X^k)}{\partial E^k} \right|_{\boldsymbol{\alpha}^k}$ (notice the additional $1/N$ term since we are using a normalized total loss function now). However, as $\left. \frac{\partial(\boldsymbol{\alpha}^k \odot X^k)}{\partial E^k} \right|_{\boldsymbol{\alpha}^k}$ is completely defined given $E^k, \boldsymbol{\alpha}^k$ and $X^{k-1}$ and by the above argument, $X^{k-1}$ never has a stale value. This shows that the only source of staleness in $\frac{\partial \mathcal{L}}{\partial E^k}$ is the staleness in values of the incomplete task gradient $\boldsymbol{\alpha}^k$. Similarly, it is easy to see that the only source of staleness in $\frac{\partial \mathcal{L}}{\partial W^{K+1}} = (X^K)^\top \mathbf{G}$ is the staleness in $\mathbf{G}$.

The above argument is easily mirrored for the backprop order of updates where an inductive argument similar to the one used to argue above that $X^k$ values are never stale in the inverted update variant, would show that the incomplete task gradient $\boldsymbol{\alpha}^k$ values are never stale in the backprop variant and the only source of staleness in $\frac{\partial \mathcal{L}}{\partial E^k}$ would then be the staleness in the $X^k$ values.

## C.4   STEP 2: RELATING THE BIAS IN MODEL GRADIENTS TO STALENESS

The above argument allows us to bound the bias in model gradients as a result of lazy updates. To avoid clutter, we will present the arguments with respect to the $E^k$ parameter. Similar arguments would hold for the $W^{K+1}$ parameter as well. Let $\tilde{\boldsymbol{\alpha}}^k, \boldsymbol{\alpha}^k$ denote the stale and actual values of the incomplete task gradient relevant to $E^k$. Let $\frac{\widetilde{\partial \mathcal{L}}}{\partial E^k} = \left. \frac{\partial(\tilde{\boldsymbol{\alpha}}^k \odot X^k)}{\partial E^k} \right|_{\tilde{\boldsymbol{\alpha}}^k}$ be the stale gradients used by IGLU in its inverted variant to update $E^k$ and similarly let $\frac{\partial \mathcal{L}}{\partial E^k} = \left. \frac{\partial(\boldsymbol{\alpha}^k \odot X^k)}{\partial E^k} \right|_{\boldsymbol{\alpha}^k}$ be the true gradient that could have been used had there been no staleness.

We will abuse notation and let the vectorized forms of these incomplete task gradients be denoted by the same symbols i.e. we stretch the matrix $\boldsymbol{\alpha}^k \in \mathbb{R}^{N \times d_k}$ into a long vector denoted also by $\boldsymbol{\alpha}^k \in \mathbb{R}^{N \cdot d_k}$. Let $\dim(E^k)$ denote the number of dimensions in the model parameter $E^k$ (recall that $E^k$ can be a stand-in for weight matrices, layer norm parameters etc used in layer $k$ of the GCN). Similarly, we let $Z_{jp}^k \in \mathbb{R}^{\dim(E^k)}, j \in [N], p \in [d_k]$ denote the vectorized form of the gradient $\frac{\partial X_{ip}^k}{\partial E^k}$ and let $Z^k \in \mathbb{R}^{N \cdot d_k \times \dim(E^k)}$ denote the matrix with all these vectors $Z_{jp}^k$ stacked up.

As per the above notation, it is easy to see that the vectorized form of the model gradient is given by

$$\frac{\partial \mathcal{L}}{\partial E^k} = \frac{(Z^k)^\top \boldsymbol{\alpha}^k}{N} \in \mathbb{R}^{\dim(E^k)}$$

The $1/N$ term appears since we are using a normalized total loss function. This also tells us that

$$\left\| \frac{\widetilde{\partial \mathcal{L}}}{\partial E^k} - \frac{\partial \mathcal{L}}{\partial E^k} \right\|_2 = \frac{\sqrt{(\tilde{\boldsymbol{\alpha}}^k - \boldsymbol{\alpha}^k)^\top (Z^k (Z^k)^\top)(\tilde{\boldsymbol{\alpha}}^k - \boldsymbol{\alpha}^k)}}{N} \leq \frac{\left\| \tilde{\boldsymbol{\alpha}}^k - \boldsymbol{\alpha}^k \right\|_2 \cdot \sigma_{\max}(Z^k)}{N}, \quad (2)$$

where $\sigma_{\max}(Z^k)$ is the largest singular value of the matrix $Z^k$.

## C.5   STEP 3: SMOOTHNESS AND BOUNDED GRADIENTS TO BOUND GRADIENT BIAS

The above discussion shows how to bound the bias in gradients in terms of staleness in the incomplete task gradients. However, to utilize this relation, we assume that the loss function $\mathcal{L}$ is $H$-smooth which is a standard assumption in literature. We will also assume that the network offers bounded gradients. Specifically, for all values of model parameters $E^k, k \in [K], W^{K+1}$ we have $\|\mathbf{G}\|_2, \left\| \frac{\partial X_{ip}^k}{\partial E^k} \right\|_2, \left\| \frac{\partial \mathcal{L}}{\partial E^k} \right\|_2, \left\| \frac{\partial X^{k+1}}{\partial X^k} \right\|_2, \|\boldsymbol{\alpha}^k\|_2 \leq B$. For sake of simplicity, we will also assume the same bound on parameters e.g. $\|\mathbf{W}^K\|_2 \leq B$. Assuming bounded gradients and bounded parameters is also standard in literature. However, whereas works such as Chen et al. (2018a) assume bounds on the sup-norm i.e. $L_\infty$ norm of the gradients, our proofs only require an $L_2$ norm bound.

We will now show that if the model parameters $\tilde{E}^k, k \in [K], \tilde{W}^{K+1}$ undergo gradient updates to their new values $E^k, k \in [K], W^{K+1}$ and the amount of *travel* is bounded by $r > 0$ i.e. $\left\| \tilde{E}^k - E^k \right\|_2 \leq r$, then we have $\left\| \tilde{\boldsymbol{\alpha}}^k - \boldsymbol{\alpha}^k \right\|_2 \leq I_k \cdot r$ where $\tilde{\boldsymbol{\alpha}}^k, \boldsymbol{\alpha}^k$ are the incomplete task gradients corresponding to

respectively old and new model parameter values and $I_k$ depends on various quantities such as the smoothness parameter and the number of layers in the network.

Lemma 1 (part 2) tells us that for the final layer, we have $\boldsymbol{\alpha}^K = \mathbf{G}(W^{K+1})^\top$ as well as for any $k < K$, we have $\boldsymbol{\alpha}^k = \left.\frac{\partial(\boldsymbol{\alpha}^{k+1} \odot X^{k+1})}{\partial X^k}\right|_{\boldsymbol{\alpha}^{k+1}}$. We analyze this using an inductive argument.

1. Case 1: $k = K$ (Base Case): In this case we have

$$\begin{aligned}
\tilde{\boldsymbol{\alpha}}^K - \boldsymbol{\alpha}^K &= \tilde{\mathbf{G}}(\tilde{W}^{K+1})^\top - \mathbf{G}(W^{K+1})^\top \\
&= (\tilde{\mathbf{G}} - \mathbf{G})(\tilde{W}^{K+1})^\top + \mathbf{G}(\tilde{W}^{K+1} - W^{K+1})^\top
\end{aligned}$$

Now, the travel condition tells us that $\left\|\tilde{W}^{K+1} - W^{K+1}\right\|_2 \le r$, boundedness tells us that $\|\mathbf{G}\|_2, \left\|\tilde{\mathbf{W}}^{K+1}\right\|_2 \le B$. Also, since the loss function is $H$-smooth, the task gradients are $H$-Lipschitz which implies along with the travel condition that $\left\|\tilde{\mathbf{G}} - \mathbf{G}\right\|_2 \le H \cdot r$. Put together this tells us that

$$\left\|\tilde{\boldsymbol{\alpha}}^K - \boldsymbol{\alpha}^K\right\|_2 \le (H+1)B \cdot r,$$

telling us that $I_K \le (H+1)B$.

2. Case 2: $k < K$ (Inductive Case): In this case, let $\tilde{X}^k$ and $X^k$ denote the embeddings with respect to the old and new parameters respectively. Then we have

$$\begin{aligned}
\tilde{\boldsymbol{\alpha}}^k - \boldsymbol{\alpha}^k &= \left.\frac{\partial(\tilde{\boldsymbol{\alpha}}^{k+1} \odot \tilde{X}^{k+1})}{\partial \tilde{X}^k}\right|_{\tilde{\boldsymbol{\alpha}}^{k+1}} - \left.\frac{\partial(\boldsymbol{\alpha}^{k+1} \odot X^{k+1})}{\partial X^k}\right|_{\boldsymbol{\alpha}^{k+1}} \\
&= \underbrace{\left.\frac{\partial(\tilde{\boldsymbol{\alpha}}^{k+1} \odot \tilde{X}^{k+1})}{\partial \tilde{X}^k}\right|_{\tilde{\boldsymbol{\alpha}}^{k+1}} - \left.\frac{\partial(\boldsymbol{\alpha}^{k+1} \odot \tilde{X}^{k+1})}{\partial \tilde{X}^k}\right|_{\boldsymbol{\alpha}^{k+1}}}_{(P)} \\
&\quad + \underbrace{\left.\frac{\partial(\boldsymbol{\alpha}^{k+1} \odot \tilde{X}^{k+1})}{\partial \tilde{X}^k}\right|_{\boldsymbol{\alpha}^{k+1}} - \left.\frac{\partial(\boldsymbol{\alpha}^{k+1} \odot X^{k+1})}{\partial X^k}\right|_{\boldsymbol{\alpha}^{k+1}}}_{(Q)}
\end{aligned}$$

By induction we have $\left\|\tilde{\boldsymbol{\alpha}}^{k+1} - \boldsymbol{\alpha}^{k+1}\right\|_2 \le I_{k+1} \cdot r$ and bounded gradients tells us $\left\|\frac{\partial \tilde{X}^{k+1}}{\partial \tilde{X}^k}\right\|_2 \le B$ giving us $\|(P)\|_2 \le I_{k+1}B \cdot r$. To analyze the term $\|(Q)\|_2$, recall that we have $X^{k+1} = f(X^k; E^{k+1})$ and since the overall task loss is $H$ smooth, so must be the function $f$. Abusing notation to let $H$ denote the Lipschitz constant of the network gives us $\left\|\tilde{X}^k - X^k\right\|_2 \le H \cdot r$. Then we have

$$\begin{aligned}
\frac{\partial(\tilde{X}^{k+1})}{\partial \tilde{X}^k} - \frac{\partial X^{k+1}}{\partial X^k} &= f'(\tilde{X}^k; \tilde{E}^{k+1}) - f'(X^k; E^{k+1}) \\
&= \underbrace{f'(\tilde{X}^k; \tilde{E}^{k+1}) - f'(X^k; \tilde{E}^{k+1})}_{(M)} + \underbrace{f'(X^k; \tilde{E}^{k+1}) - f'(X^k; E^{k+1})}_{(N)}
\end{aligned}$$

Applying smoothness, we have $\|(M)\|_2 \le H^2 \cdot r$ as well as $\|(N)\|_2 \le H \cdot r$. Together with bounded gradients that gives us $\left\|\boldsymbol{\alpha}^{k+1}\right\|_2 \le B$, we have $\|(Q)\|_2 \le BH(H+1) \cdot r$. Together, we have

$$\left\|\tilde{\boldsymbol{\alpha}}^k - \boldsymbol{\alpha}^k\right\|_2 \le B(H(H+1) + I_{k+1}) \cdot r,$$

telling us that $I_k \le B(H(H+1) + I_{k+1})$.

The above tells us that in Algorithm 2, suppose the parameter update steps i.e. step 7 and step 10 are executed by effecting $\tau$ mini-batch SGD steps or else $\tau$ full-batch GD steps each time, with

step length $\eta$, then the amount of travel in any model parameter is bounded above by $\tau\eta B$ i.e. $\left\|\tilde{E}^k - E^k\right\|_2 \leq \tau\eta B$ and so on. Now, incomplete task gradients $\boldsymbol{\alpha}^k$ are updated only after model parameters for all layers have been updated once and the algorithm loops back to step 6. Thus, Lipschitzness of the gradients tells us that the staleness in the incomplete task gradients, for any $k \in [K]$, is upper bounded by

$$\left\|\tilde{\boldsymbol{\alpha}}^k - \boldsymbol{\alpha}^k\right\|_2 \leq \tau\eta \cdot IB,$$

where we take $I = \max_{k \leq K} I_k$ for sake of simplicity. Now, since $\left\|\frac{\partial X_{i_p}^k}{\partial E^k}\right\|_2 \leq B$ as gradients are bounded, we have $\sigma_{\max}(Z^k) \leq N \cdot d_k B$. Then, combining with the result in Equation (2) gives us

$$\left\|\frac{\widetilde{\partial \mathcal{L}}}{\partial E^k} - \frac{\partial \mathcal{L}}{\partial E^k}\right\|_2 \leq \tau\eta \cdot IB^2 d_k$$

Taking in contributions of gradients of all layers and the final classifier layer gives us the bias in the total gradient using triangle inequality as

$$\left\|\widetilde{\nabla\mathcal{L}} - \nabla\mathcal{L}\right\|_2 \leq \sum_{k \in [K]} \left\|\frac{\widetilde{\partial \mathcal{L}}}{\partial E^k} - \frac{\partial \mathcal{L}}{\partial E^k}\right\|_2 + \left\|\frac{\widetilde{\partial \mathcal{L}}}{\partial W^{K+1}} - \frac{\partial \mathcal{L}}{\partial W^{K+1}}\right\|_2 \leq \tau\eta \cdot IB^2 d_{\max}(K+1),$$

where $d_{\max} = \max_{k \in K} d_k$ is the maximum embedding dimensionality of any layer.

## C.6  STEP 4 (I): CONVERGENCE GUARANTEE (MINI-BATCH SGD)

Let us analyze convergence in the case when updates are made using mini-batch SGD in steps 7, 10 of Algorithm 2. The discussion above establishes an upper bound on the *absolute* bias in the gradients. However, our proofs later require a *relative* bound which we tackle now. Let us decide to set the step length to $\eta = \frac{1}{C\sqrt{T}}$ for some constant $C > 0$ that will be decided later and also set some value $\phi < 1$. Then two cases arise

1.  **Case 1**: The relative gradient bias is too large i.e. $\tau\eta \cdot IB^2 d_{\max}(K+1) > \phi \cdot \|\nabla\mathcal{L}\|_2$. In this case we are actually done since we get

    $$\|\nabla\mathcal{L}\|_2 \leq \frac{\tau \cdot IB^2 d_{\max}(K+1)}{\phi \cdot C\sqrt{T}},$$

    i.e. we are already at an approximate first-order stationary point.

2.  **Case 2**: The relative gradient bias is small i.e. $\tau\eta \cdot IB^2 d_{\max}(K+1) \leq \phi \cdot \|\nabla\mathcal{L}\|_2$. In this case we satisfy the relative bias bound required by Lemma 5 (part 2) with $\delta = \phi$.

This shows that either Case 1 happens in which case we are done or else Case 2 keeps applying which means that Lemma 5 (part 2) keeps getting its prerequisites satisfied. If Case 1 does not happen for $T$ steps, then Lemma 5 (part 2) assures us that we will arrive at a point where

$$\mathbb{E}\left[\|\nabla\mathcal{L}\|_2^2\right] \leq \frac{2C(\mathcal{L}^0 - \mathcal{L}^*) + H\sigma^2/C}{(1-\phi)\sqrt{T}}$$

where $\mathcal{L}^0, \mathcal{L}^*$ are respectively the initial and optimal values of the loss function which we recall is $H$-smooth and $\sigma^2$ is the variance due to mini-batch creation and we set $\eta = \frac{1}{C\sqrt{T}}$. Setting $C = \sqrt{\frac{H\sigma^2}{2(\mathcal{L}^0 - \mathcal{L}^*)}}$ tells us that within $T$ steps, either we will achieve

$$\|\nabla\mathcal{L}\|_2^2 \leq \frac{2\tau^2 \cdot I^2 B^4 d_{\max}^2 (K+1)^2(\mathcal{L}^0 - \mathcal{L}^*)}{\phi^2 H\sigma^2 T},$$

or else we will achieve

$$\mathbb{E}\left[\|\nabla\mathcal{L}\|_2^2\right] \leq \frac{2\sigma\sqrt{2(\mathcal{L}^0 - \mathcal{L}^*)H}}{(1-\phi)\sqrt{T}}$$

Setting $\tau = T^{\frac{1}{4}} \cdot \left( \frac{\sqrt{\sigma^3 H \sqrt{H}}}{(\mathcal{L}^0 - \mathcal{L}^*)^{\frac{1}{4}} I B^2 d_{\max}(K+1)} \right)$ balances the two quantities in terms of their dependence on $T$ (module absolute constants such as $\phi$). Note that as expected, as the quantities $I, B, d_{\max}, K$ increase, the above limit on $\tau$ goes down i.e. we are able to perform fewer and fewer updates to the model parameters before a refresh is required. This concludes the proof of Theorem 4 for the second case.

### C.7 STEP 4 (II): CONVERGENCE GUARANTEE (FULL-BATCH GD)

In this case we similarly have either the relative gradient bias to be too large in which case we get

$$\|\nabla \mathcal{L}\|_2 \leq \frac{\tau \eta \cdot I B^2 d_{\max}(K+1)}{\phi},$$

or else we satisfy the relative bias bound required by Lemma 5 (part 1) with $\delta = \phi$. This shows that either Case 1 happens in which case we are done or else Case 2 keeps applying which means that Lemma 5 (part 1) keeps getting its prerequisites satisfied. If Case 1 does not happen for $T$ steps, then Lemma 5 (part 1) assures us that we will arrive at a point where

$$\|\nabla \mathcal{L}\|_2^2 \leq \frac{2(\mathcal{L}^0 - \mathcal{L}^*)}{\eta(1-\phi)T}$$

In this case, for a given value of $\tau$, setting $\eta = \left( \frac{2(\mathcal{L}^0 - \mathcal{L}^*)}{(1-\phi)\tau^2 I^2 B^4 d_{\max}^2 (K+1)^2 T} \right)^{\frac{1}{3}}$ gives us that within $T$ iterations, we must achieve

$$\|\nabla \mathcal{L}\|_2^2 \leq \left( \frac{2\tau(\mathcal{L}^0 - \mathcal{L}^*) I B^2 d_{\max}(K+1)}{(1-\phi)} \right)^{\frac{2}{3}} \cdot \frac{1}{T^{\frac{2}{3}}}$$

In this case, it is prudent to set $\tau = \mathcal{O}(1)$ so as to not deteriorate the convergence rate. This concludes the proof of Theorem 4 for the first case.

### C.8 GENERIC CONVERGENCE RESULTS

**Lemma 5** (First-order Stationarity with a Smooth Objective). *Let $f : \boldsymbol{\Theta} \to \mathbb{R}$ be an $H$-smooth objective over model parameters $\theta \in \boldsymbol{\Theta}$ that is being optimized using a gradient oracle and the following update for step length $\eta$:*

$$\boldsymbol{\theta}^{t+1} = \boldsymbol{\theta}^t - \eta \cdot \mathbf{g}^t$$

*Let $\boldsymbol{\theta}^*$ be an optimal point i.e. $\boldsymbol{\theta}^* \in \arg\min_{\boldsymbol{\theta} \in \boldsymbol{\Theta}} f(\boldsymbol{\theta})$. Then, the following results hold depending on the nature of the gradient oracle:*

1. *If a non-stochastic gradient oracle with bounded bias is used i.e. for some $\delta \in (0, 1)$, for all $t$, we have $\mathbf{g}^t = \nabla f(\boldsymbol{\theta}^t) + \boldsymbol{\Delta}^t$ where $\|\boldsymbol{\Delta}^t\|_2 \leq \delta \cdot \|\nabla f(\boldsymbol{\theta}^t)\|_2$ and if the step length satisfies $\eta \leq \frac{(1-\delta)}{2H(1+\delta^2)}$, then for any $T > 0$, for some $t \leq T$ we must have*

$$\left\| \nabla f(\boldsymbol{\theta}^t) \right\|_2^2 \leq \frac{2(f(\boldsymbol{\theta}^0) - f(\boldsymbol{\theta}^*))}{\eta(1-\delta)T}$$

2. *If a stochastic gradient oracle is used with bounded bias i.e. for some $\delta \in (0, 1)$, for all $t$, we have $\mathbb{E}\left[ \mid \mathbf{g}^t \boldsymbol{\theta}^t \right] = \nabla f(\boldsymbol{\theta}^t) + \boldsymbol{\Delta}^t$ where $\|\boldsymbol{\Delta}^t\|_2 \leq \delta \cdot \|\nabla f(\boldsymbol{\theta}^t)\|_2$, as well as bounded variance i.e. for all $t$, we have $\mathbb{E}\left[ \mid \|\mathbf{g}^t - \nabla f(\boldsymbol{\theta}^t) - \boldsymbol{\Delta}^t\|_2^2 \boldsymbol{\theta}^t \right] \leq \sigma^2$ and if the step length satisfies $\eta \leq \frac{(1-\delta)}{2H(1+\delta^2)}$, as well as $\eta = \frac{1}{C\sqrt{T}}$ for some $C > 0$, then for any $T > \frac{4H^2(1+\delta^2)^2}{C^2(1-\delta)^2}$, for some $t \leq T$ we must have*

$$\mathbb{E}\left[ \left\| \nabla f(\boldsymbol{\theta}^t) \right\|_2^2 \right] \leq \frac{2C(f(\boldsymbol{\theta}^0) - f(\boldsymbol{\theta}^*)) + H\sigma^2/C}{(1-\delta)\sqrt{T}}$$

*Proof (of Lemma 5).* To prove the first part, we notice that smoothness of the objective gives us

$$f(\boldsymbol{\theta}^{t+1}) \le f(\boldsymbol{\theta}^t) + \langle \nabla f(\boldsymbol{\theta}^t), \boldsymbol{\theta}^{t+1} - \boldsymbol{\theta}^t \rangle + \frac{H}{2} \left\| \boldsymbol{\theta}^{t+1} - \boldsymbol{\theta}^t \right\|_2^2$$

Since we used the update $\boldsymbol{\theta}^{t+1} = \boldsymbol{\theta}^t - \eta \cdot \mathbf{g}^t$ and we have $\mathbf{g}^t = \nabla f(\boldsymbol{\theta}^t) + \boldsymbol{\Delta}^t$, the above gives us

$$f(\boldsymbol{\theta}^{t+1}) \le f(\boldsymbol{\theta}^t) - \eta \cdot \langle \nabla f(\boldsymbol{\theta}^t), \mathbf{g}^t \rangle + \frac{H\eta^2}{2} \left\| \mathbf{g}^t \right\|_2^2$$

$$= f(\boldsymbol{\theta}^t) - \eta \cdot \langle \nabla f(\boldsymbol{\theta}^t), \nabla f(\boldsymbol{\theta}^t) + \boldsymbol{\Delta}^t \rangle + \frac{H\eta^2}{2} \left( \left\| \nabla f(\boldsymbol{\theta}^t) + \boldsymbol{\Delta}^t \right\|_2^2 \right)$$

$$= f(\boldsymbol{\theta}^t) - \eta \cdot \left\| \nabla f(\boldsymbol{\theta}^t) \right\|_2^2 - \eta \cdot \langle \nabla f(\boldsymbol{\theta}^t), \boldsymbol{\Delta}^t \rangle + \frac{H\eta^2}{2} \left( \left\| \nabla f(\boldsymbol{\theta}^t) + \boldsymbol{\Delta}^t \right\|_2^2 \right)$$

Now, the Cauchy-Schwartz inequality along with the bound on the bias gives us $-\eta \cdot \langle \nabla f(\boldsymbol{\theta}^t), \boldsymbol{\Delta}^t \rangle \le \eta\delta \cdot \left\| \nabla f(\boldsymbol{\theta}^t) \right\|_2^2$ as well as $\left\| \nabla f(\boldsymbol{\theta}^t) + \boldsymbol{\Delta}^t \right\|_2^2 \le 2(1 + \delta^2) \cdot \left\| \nabla f(\boldsymbol{\theta}^t) \right\|_2^2$. Using these gives us

$$f(\boldsymbol{\theta}^{t+1}) \le f(\boldsymbol{\theta}^t) - \eta(1 - \delta - \eta H(1 + \delta^2)) \cdot \left\| \nabla f(\boldsymbol{\theta}^t) \right\|_2^2$$

$$\le f(\boldsymbol{\theta}^t) - \frac{\eta(1 - \delta)}{2} \cdot \left\| \nabla f(\boldsymbol{\theta}^t) \right\|_2^2,$$

since we chose $\eta \le \frac{(1-\delta)}{2H(1+\delta^2)}$. Reorganizing, taking a telescopic sum over all $t$, using $f(\boldsymbol{\theta}^{T+1}) \ge f(\boldsymbol{\theta}^*)$ and making an averaging argument tells us that since we set , for any $T > 0$, it must be the case that for some $t \le T$, we have

$$\left\| \nabla f(\boldsymbol{\theta}^t) \right\|_2^2 \le \frac{2(f(\boldsymbol{\theta}^0) - f(\boldsymbol{\theta}^*))}{\eta(1 - \delta)T}$$

This proves the first part. For the second part, we yet again invoke smoothness to get

$$f(\boldsymbol{\theta}^{t+1}) \le f(\boldsymbol{\theta}^t) + \langle \nabla f(\boldsymbol{\theta}^t), \boldsymbol{\theta}^{t+1} - \boldsymbol{\theta}^t \rangle + \frac{H}{2} \left\| \boldsymbol{\theta}^{t+1} - \boldsymbol{\theta}^t \right\|_2^2$$

Since we used the update $\boldsymbol{\theta}^{t+1} = \boldsymbol{\theta}^t - \eta \cdot \mathbf{g}^t$, the above gives us

$$f(\boldsymbol{\theta}^{t+1}) \le f(\boldsymbol{\theta}^t) - \eta \cdot \langle \nabla f(\boldsymbol{\theta}^t), \mathbf{g}^t \rangle + \frac{H\eta^2}{2} \left\| \mathbf{g}^t \right\|_2^2$$

$$= f(\boldsymbol{\theta}^t) - \eta \cdot \langle \nabla f(\boldsymbol{\theta}^t), \mathbf{g}^t \rangle + \frac{H\eta^2}{2} \left( \left\| \nabla f(\boldsymbol{\theta}^t) + \boldsymbol{\Delta}^t \right\|_2^2 + \left\| \mathbf{g}^t - \nabla f(\boldsymbol{\theta}^t) - \boldsymbol{\Delta}^t \right\|_2^2 \right)$$

$$- H\eta^2 \langle \nabla f(\boldsymbol{\theta}^t) + \boldsymbol{\Delta}^t, \mathbf{g}^t - \nabla f(\boldsymbol{\theta}^t) - \boldsymbol{\Delta}^t \rangle$$

Taking conditional expectations on both sides gives us

$$\mathbb{E}\left[ \, | \, f(\boldsymbol{\theta}^{t+1}) \boldsymbol{\theta}^t \right] \le f(\boldsymbol{\theta}^t) - \eta \cdot \left\| \nabla f(\boldsymbol{\theta}^t) \right\|_2^2 - \eta \cdot \langle \nabla f(\boldsymbol{\theta}^t), \boldsymbol{\Delta}^t \rangle + \frac{H\eta^2}{2} \left( \left\| \nabla f(\boldsymbol{\theta}^t) + \boldsymbol{\Delta}^t \right\|_2^2 + \sigma^2 \right)$$

Now, the Cauchy-Schwartz inequality along with the bound on the bias gives us $-\eta \cdot \langle \nabla f(\boldsymbol{\theta}^t), \boldsymbol{\Delta}^t \rangle \le \eta\delta \cdot \left\| \nabla f(\boldsymbol{\theta}^t) \right\|_2^2$ as well as $\left\| \nabla f(\boldsymbol{\theta}^t) + \boldsymbol{\Delta}^t \right\|_2^2 \le 2(1 + \delta^2) \cdot \left\| \nabla f(\boldsymbol{\theta}^t) \right\|_2^2$. Using these and applying a total expectation gives us

$$\mathbb{E}\left[ f(\boldsymbol{\theta}^{t+1}) \right] \le \mathbb{E}\left[ f(\boldsymbol{\theta}^t) \right] - \eta(1 - \delta - \eta H(1 + \delta^2)) \cdot \mathbb{E}\left[ \left\| \nabla f(\boldsymbol{\theta}^t) \right\|_2^2 \right] + \frac{H\eta^2}{2} \cdot \sigma^2$$

$$\le \mathbb{E}\left[ f(\boldsymbol{\theta}^t) \right] - \frac{\eta(1 - \delta)}{2} \cdot \mathbb{E}\left[ \left\| \nabla f(\boldsymbol{\theta}^t) \right\|_2^2 \right] + \frac{H\eta^2}{2} \cdot \sigma^2$$

where the second step follows since we set $\eta \le \frac{(1-\delta)}{2H(1+\delta^2)}$. Reorganizing, taking a telescopic sum over all $t$, using $f(\boldsymbol{\theta}^{T+1}) \ge f(\boldsymbol{\theta}^*)$ and making an averaging argument tells us that for any $T > 0$, it must be the case that for some $t \le T$, we have

$$\mathbb{E}\left[ \left\| \nabla f(\boldsymbol{\theta}^t) \right\|_2^2 \right] \le \frac{2(f(\boldsymbol{\theta}^0) - f(\boldsymbol{\theta}^*))}{\eta(1 - \delta)T} + \frac{H\eta\sigma^2}{1 - \delta}$$

However, since we also took care to set $\eta = \frac{1}{C\sqrt{T}}$, we get

$$\mathbb{E}\left[ \left\| \nabla f(\boldsymbol{\theta}^t) \right\|_2^2 \right] \le \frac{2C(f(\boldsymbol{\theta}^0) - f(\boldsymbol{\theta}^*)) + H\sigma^2/C}{(1 - \delta)\sqrt{T}}$$

which proves the second part and finishes the proof. $\square$

