# OpenReview forum: "IGLU: Efficient GCN Training via Lazy Updates"
_ICLR.cc/2022/Conference — ICLR 2022 Poster_

### Official Review · Reviewer_jiPr · 2021-10-31

**Correctness:** 4
**Technical Novelty And Significance:** 3
**Empirical Novelty And Significance:** 3
**Recommendation:** 6
**Confidence:** 3

**Main Review:**

Overall, IGLU represents a novel method for achieving efficient GNN training. IGLU's idea is quite different from the common scalable GNN training methods, such as GraphSAGE and ClusterGCN.
The authors provide theories to show that the bias of IGLU is bounded.
The experimental results are promising as well.

Main questions:
1. The overall algorithms are still vague for me. I can understand the main results that we can use the previous layer's embeddings to approximate the exact backpropagated gradients.
I appreciate the authors presenting Algorithm 1 and 2 as well. However, the exact computation is vague. For example, in the algorithms it said things like "Compute G", "using Lemma 1 (1)", but it's unclear how they are used.

2. The evaluation setup is not rigorous. In Table 1, different GNN architectures are used to produce the results, which makes things unclear how much gain IGLU training is offering.
I believe to show that IGLU is really model agnostic, at least one set of experiments should be made, where everything about GNN architecture is fixed, but only the training algorithms vary.
For example, you may refer to the paper "Design Space for Graph Neural Networks" on rigorously comparing different GNN design options.
I believe only with rigorous controlled experiments, the results in Table 1 Figure 2 can assure the readers of the effectiveness of IGLU.

3. In Table 1, the percentage of speedup is quite significant. Could the authors explain how they are computed? What is the baseline speed?

4. Could the authors explain the rationale for IGLU outperforming full-batch GNN training? I believe full-batch training can offer exact gradients.

5. In Figure 2, the results on "OGBN-Proteins" seem suspicious. It seems that without training, IGLU can already achieve much higher ROC AUC over other methods. I hope the authors explain the reasons behind it.




**Summary Of The Paper:**

This paper introduces a new method, IGLU, that caches intermediate computations at various GCN layers.
This enables IGLU to perform lazy updates that do not require updating a large number of node embeddings during descent and offers much faster convergence without significantly biasing the gradients.

Overall, this paper represents a novel solution towards efficient GNN training.

**Summary Of The Review:**

Overall, I like the technical contributions made in this paper.
My concerns are:
1. The algorithms are not clearly described, making it hard to understand the implementation of IGLU
2. The evaluation setup is not rigorous, making the claims on the effectiveness of IGLU questionable

---

> ### Author Response · Authors · 2021-11-17
> **Response to Reviewer jiPr (Part 1)**
>
> We thank the reviewer for insightful comments and suggestions. We have summarized your concerns and provided responses for the same below.
>
> > **The overall algorithms are still vague for me. I can understand the main results that we can use the previous layer's embeddings to approximate the exact backpropagated gradients. I appreciate the authors presenting Algorithm 1 and 2 as well. However, the exact computation is vague. For example, in the algorithms it said things like "Compute G", "using Lemma 1 (1)", but it's unclear how they are used.**
>
> We regret that the concise presentation of the algorithms in the main paper due to lack of space was not entirely clear. We have included additional content to the paper in Section A.7 of the Appendix (Page 20-22)  to give expanded versions of the algorithms which hopefully give all the details. In summary, we would like to point out that Lemma~1.1 shows that the exact backprop gradients can be computed using two sets of quantities, namely the incomplete task gradients $\alpha^k, k \in [K]$ and the layerwise embeddings $X^k, k \in [K]$. The backprop variant of IGLU (Algorithm 1) uses stale $X^k, k \in [K]$ and freshly computed $\alpha^k, k \in [K]$ whereas the inverted variant of IGLU (Algorithm 2) uses freshly computed $X^k, k \in [K]$ but stale $\alpha^k, k \in [K]$. It is notable that, as pointed out in page 5 of the main paper (please see "Contributions" paragraph), previous work seems to focus only on stale embeddings (as done by Algorithm 1) and does not explore options such as those offered by Algorithm 2 that keep incomplete gradients stale instead. Interestingly, in experiments, it is Algorithm 2 that outperforms Algorithm 1.
>
> ---
>
> >**The evaluation setup is not rigorous. In Table 1, different GNN architectures are used to produce the results, which makes things unclear how much gain IGLU training is offering. I believe to show that IGLU is really model agnostic, at least one set of experiments should be made, where everything about GNN architecture is fixed, but only the training algorithms vary. For example, you may refer to the paper "Design Space for Graph Neural Networks" on rigorously comparing different GNN design options. I believe only with rigorous controlled experiments, the results in Table 1 Figure 2 can assure the readers of the effectiveness of IGLU.**
>
> We strictly follow the evaluation setup for all the baseline methods as described in GraphSAINT [1] and use the author’s official implementations for the corresponding best results for each baseline as described in Section A.4 of the Appendix.
> Since there are multiple baseline methods and architectures for comparison, we use only two architectures for IGLU, namely the VR-GCN and GraphSAINT style architectures and offer a performance comparison with the same in Table 1 and Figure 2. We use the GraphSAINT style architecture (exactly identical to the GraphSAINT baseline’s architecture) for the PPI-Large, Flickr and OGBN-Proteins datasets and VR-GCN style architecture (exactly identical to the VR-GCN baseline’s architecture) for the Reddit and OGBN-Arxiv datasets (with parameters such as Number of Layers, Hidden Dimensions, etc fixed and uniform across architectures) to clearly demonstrate the improvement in performance and convergence that IGLU offers.
>
> **Results**: To summarize, IGLU with GraphSAINT architecture offers a relative improvement of 3.24% on PPI-Large, 0.98% on Flickr and 2.61% on Proteins, along with significant speedups, as compared to the GraphSAINT baseline with the same architecture. Similarly, IGLU with VR-GCN style architecture offers a relative improvement of 2.42% on the Arxiv dataset and competitive performance with significant speedups on the Reddit dataset as compared to the VR-GCN baseline with the same architecture.
>
> Additionally, we also provide experiments with GAT, GCN and GraphSAGE architectures and their comparison with identical baseline architectures in Section B.6 of the Appendix to demonstrate IGLU’s effectiveness.
>
> [1] GraphSAINT: Graph Sampling Based Inductive Learning Method, Zeng et. al, ICLR ‘20

---

> > ### Author Response · Authors · 2021-11-17
> > **Response to Reviewer jiPr (Part 2)**
> >
> > Response to Reviewer jiPr (continued..)
> >
> > >**In Table 1, the percentage of speedup is quite significant. Could the authors explain how they are computed? What is the baseline speed?**
> >
> > We compute the percentage speedup that IGLU achieves as a relative measure of time taken by IGLU to reach the highest validation performance of the best baseline. Mathematically, we describe the % speedup as Equation (1) on Page 7 of the paper: let the highest validation score obtained by the best baseline be $v_{1}$ and $t_{1}$ be the time taken to reach that score. Let the time taken by IGLU to reach $v_{1}$ be $t_{2}$. Then, we define the percentage speedup as,
> > % Speedup $:= \frac{t_{1} - t_{2}}{t_{1}} \times 100$
> >
> > We also provide the training times for the best performing baseline and for IGLU to reach the validation score $v_{1}$, along with the percentage speedup values computed using the above formula, in the table below for clarity.
> >
> > **Table:** Speedup computation with training times the best baseline and IGLU
> >
> > | **Algorithm**    | **PPI-Large** | **Reddit** | **Flickr** | **Proteins** | **Arxiv** |
> > | ----------- | ----------- |----------- |----------- |----------- |----------- |
> > | Validation Score ($v_{1}$) | 0.961 | 0.955* | 0.511 | 0.855 | 0.721 |
> > | Best Baseline | VR-GCN | GraphSAINT  | GraphSAINT | GraphSAINT | GraphSAINT |
> > | Time: Baseline ($t_{1}$) | 368.4 | 2.74 | 1.43 | 599.1 | 44.6 |
> > | Time: IGLU ($t_{2}$) | 43.7 | 2.52  | 0.79 | 532.8 | 38.4 |
> > | **% Speedup (Eq. 1)**   | **88.12**    | **8.1**   | **44.74** | **11.05** | **13.94** |
> >
> > $*$ For Reddit, speedup is computed at a *high* validation score of 0.955.
> >
> > ---
> >
> > >**Could the authors explain the rationale for IGLU outperforming full-batch GNN training? I believe full-batch training can offer exact gradients.**
> >
> > We would like to clarify that the performance numbers reported for GCN in the main paper are for a mini-batch training methodology, strictly following the evaluation setup in GraphSAINT and is not full-batch training. We apologise for any confusion this omission might have caused and have updated this detail in the revised version.
> >
> > ---
> >
> > >**In Figure 2, the results on "OGBN-Proteins" seem suspicious. It seems that without training, IGLU can already achieve much higher ROC AUC over other methods. I hope the authors explain the reasons behind it.**
> >
> > We would like to clarify that in Figure 2, the performance is measured at the granularity of an epoch, hence the first datapoint corresponds to the performance at the end of the first epoch. We apologise for any confusions caused in this regard.
> > To further provide a fine-grained analysis of IGLU’s fast convergence on the OGBN-Proteins dataset, we measure the Validation ROC-AUC at the granularity of a minibatch for the first epoch and discuss the results of the analysis in Section A.8 (Page 22) of the Appendix.
> >
> > We plot the performance as a function of the number of mini batches in Figure 7 (page 23 in Section A.8 of the Appendix) for the first epoch of training. Since the layers in IGLU are trained sequentially, we observe that the validation ROC AUC begins at a value of 0.5 at the beginning of training, rises to roughly 0.51 once the layer closest to the input is trained, increases to roughly 0.57 once the subsequent layer is trained and eventually reaches a high validation ROC AUC close to 0.81 once the layer closest to the output is trained. In Figure 2 of the main paper, the high validation score reflects this result at the end of the first epoch. Training for this first epoch approximately takes 5 seconds as depicted in Figure 2.

---

> ### Author Response · Authors · 2021-11-19
> **Follow-up on response to Reviewer jiPr**
>
> Dear Reviewer jiPr,
>
> We would like to thank you for your insightful comments and suggestions that have significantly improved the paper.
> We hope that our responses and revision address your concerns satisfactorily and will lead to a more positive evaluation of our work. With the discussion window closing soon, we wanted to know if there are any further clarifications that we can provide from our end.

---

> > ### Author Response · Authors · 2021-11-30
> > **Gentle follow-up to Reviewer jiPr**
> >
> > Dear Reviewer jiPr,
> >
> > We would like to thank you for the constructive feedback on our paper. We have tried to incorporate the feedback in the latest draft, hopefully this positively impacts your opinion of the paper.
> >
> > Gentle follow up on the responses and if there are any further questions/concerns we can address with the discussion window closing in a couple of hours.

---

### Official Review · Reviewer_mKbp · 2021-11-02

**Correctness:** 3
**Technical Novelty And Significance:** 3
**Empirical Novelty And Significance:** 3
**Recommendation:** 6
**Confidence:** 3

**Main Review:**

Strengths:
1. The scalability issue of GNNs is challenging and an important research topic. IGLU is simple and the motivation is clear and intuitive.
2. Compared to strong baselines, IGLU has better empirical performance in both accuracy and training time.
3. Under certain assumptions, IGLU has non-trivial convergence guarantees, which is an advantage against baseline approaches.


Weaknesses:
1. Caching intermediate results to speed up GNN training has appeared in multiple prior works. The main difference is that this paper considers two variants: cache embeddings or gradients. However, I still think the technical novelty of IGLU is not significant.
2. The GNNAutoScale method of Fey et al. uses similar ideas and make uses of stale embeddings to scale up the training of GNNs. From their experimental results, GNNAutoScale achieves much better performance compared to GraphSAITNT, Cluster-GCN, etc.  So, I think the authors should provides a more detailed discussion on the difference and connection between IGLU and GNNAutoScale. Moreover, empirical comparison between them would make the results more complete and convincing.


**Summary Of The Paper:**

This paper studies tries to tackle the scalability challenge of training GNNs on large graphs. The authors propose IGLU, an architecture-agnostic method. IGLU caches intermediate computations and uses a lazy update strategy. Convergence analysis on IGLU is provided and empirical results show that IGLU has better performance than baselines such as GraphSAINT, Cluster-GCN, VR-GCN.


**Summary Of The Review:**

Overall, this paper has some insights on tackling the scalability challenge training GNNs on large graphs. And IGLU is an effective method for scaling up GNN training and shows better empirical performance compared to well-known techniques. On the downside, the technical novelty is not significant. Moreover, I think authors should provide a more detailed comparison between IGLU and GNNAutoScale.

---

> ### Author Response · Authors · 2021-11-17
> **Response to Reviewer mKbp (Part 1)**
>
> We thank the reviewer for insightful comments and suggestions. We have summarized your concerns and provided responses for the same below.
>
> > **Caching intermediate results to speed up GNN training has appeared in multiple prior works. The main difference is that this paper considers two variants: cache embeddings or gradients. However, I still think the technical novelty of IGLU is not significant.**
>
> **Re Technical Novelty**
>
> We humbly disagree with the comment that IGLU lacks technical novelty as compared to prior works. We highlight the key difference between earlier works that cache intermediate results for speeding up GNN training namely VR-GCN, MVS-GNN and GNNAutoScale in Section A.6 of the Appendix (Page 18-21) and discuss IGLU’s technical novelty in detail as compared to these prior works.
>
> We also discuss the same below, for convenience:
>
> ### **VRGCN[1] v/s IGLU**
>
>
> 1. **Update of Cached Embeddings:** VR-GCN caches only historical embeddings and while processing a single mini-batch these historical embeddings are updated for a sampled subset of the nodes. In contrast IGLU **does not update** any intermediate results after processing each mini-batch. These are updated only **once per epoch**, after all parameters for individual layers have been updated.
>
> 2. **Update of Model Parameters**: VR-GCN’s backpropagation step involves update of model parameters of all layers after each mini-batch. In contrast IGLU updates parameters of **only a single layer at a time.**
>
> 3. **Variance Artifacts due to Sampling**: VR-GCN incurs additional variance due to neighborhood sampling which is then reduced by utilizing historical embeddings for some nodes and by computing exact embeddings for the others. IGLU does not incur such variance since IGLU uses all the neighbors.
>
>
> ### **MVS-GNN v/s IGLU**
>
> MVS-GNN [2] is another work that caches historical embeddings. It follows a nested training strategy wherein firstly a large batch of nodes are sampled and mini-batches are further created from this large batch for training. MVS-GNN handles variance due to this mini-batch creation by performing importance weighted sampling to construct mini-batches.
>
> 1. **Update of Cached Embeddings and Variance due to Sampling:** Building upon VR-GCN, to reduce the variance in embeddings due to its sampling of nodes at different layers, MVS-GNN caches only embeddings and uses historical embeddings for some nodes and recomputes the embeddings for the others. Similar to VR-GCN, these historical embeddings are updated as and when they are part of the mini-batch used for training. As discussed above, IGLU does not incur such variance since IGLU uses all the neighbors.
>
> 2. **Update of Model Parameters:** Update of model parameters in MVS-GNN is similar to that of VR-GCN, where backpropagation step involves update of model parameters of all layers for each mini-batch. As described already, IGLU updates parameters of only a single layer at a time.

---

> > ### Author Response · Authors · 2021-11-17
> > **Response to Reviewer mKbp (Part 2)**
> >
> > Response to Reviewer mKbp (continued..)
> >
> > ### **GNNAutoScale v/s IGLU**
> >
> > First, we would like to highlight that IGLU was submitted to NeurIPS 2021 in May 2021 while GNNAutoscale appeared in June 2021. So the works are concurrent and independent.
> >
> > Next, we discuss the differences between the two methods. GNNAutoScale [3] extends the idea of caching historical embeddings from VR-GCN and provides a scalable solution.
> >
> > 1. **Update of intermediate representations and model parameters:** While processing a minibatch of nodes, GNNAutoScale computes the embeddings for these nodes at each layer while using historical embeddings for the one-hop neighbors outside the current minibatch. After processing each mini-batch, GNNAutoScale updates the historical embeddings for nodes considered in the mini-batch. Similar to VR-GCN and MVS-GNN, GNNAutoScale updates all parameters at all layers while processing a mini-batch of nodes. In contrast IGLU does not update intermediate results (intermediate representations in Algorithm 1 and incomplete gradients in Algorithm 2) after processing each minibatch. In fact, these are updated only once per epoch, after all parameters for individual layers have been updated.
> >
> > 2. **Partitioning:** GNNAutoScale relies on the METIS clustering algorithm for creating mini-batches that minimize inter-connectivity across batches. This is done to minimize access to historical embeddings and reduce staleness. These algorithms tend to bring similar nodes together, potentially resulting in the distributions of clusters being different from the original dataset. This may lead to biased estimates of the full gradients while training using mini-batch SGD as has been discussed in Section 3.2 of ClusterGCN[4] (Page 5). IGLU does not rely on such algorithms and avoids potential additional bias.
> >
> >
> >
> > **Similarity of IGLU with GNNAutoScale:** Both of the methods avoid a neighborhood sampling step, thereby avoiding additional variance due to neighborhood sampling. Both IGLU and GNNAutoScale propose methods to reduce the neighborhood explosion problem although in fundamentally different manners. GNNAutoScale does so by pruning the computation graph by using historical embeddings for neighbors across different layers. IGLU on the other hand restricts the parameter updates to a single layer at a time by analyzing the gradient structure of GNNs.
> >
> > ### **Summary of IGLU's Technical Novelty and Contrast with Caching Based Related Works**
> >
> > To summarize, IGLU is fundamentally different from these methods that cache historical embeddings in that it changes the entire training procedure of GCNs in contrast with the aforementioned caching based methods as follows:
> >
> > 1. The above methods still follow standard **SGD style training of GCNs** in that they update the model parameters at all the layers after each mini-batch. This is very different from IGLU’s parameter updates that concern **only a single layer** at a time.
> > 2. IGLU can cache either **incomplete gradients OR stale embeddings** which is different to the other approaches that cache **only embeddings.** This provides alternate approaches for training GCNs and we demonstrate empirically that caching incomplete gradients, in fact, can offer superior performance and convergence.
> > 3. Unlike GNNAutoScale and VR-GCN that update some of the historical embeddings after each minibatch is processed, IGLU’s caching is **much more aggressive** and the stale variables are updated **only once per epoch**, after all parameters for all layers have been updated.
> > 4. Theoretically, we provide **good convergence rates and bounded bias** even while using **stale gradients**, which has not been discussed in any prior works.
> >
> > These are the main technical novelties of our proposed method and they are a consequence of a careful understanding of the gradient structure of GCN’s themselves.
> >
> > [1] Stochastic Training of Graph Convolutional Networks with Variance Reduction, Chen et. al, ICML ‘18
> >
> > [2] Minimal Variance Sampling with Provable Guarantees for Fast Training of Graph Neural Networks, Cong et. al, KDD ‘20
> >
> > [3] GNNAutoScale: Scalable and Expressive Graph Neural Networks via Historical Embeddings, Fey et. al, ICML ‘21
> >
> > [4] ClusterGCN: An Efficient Training Algorithm for Training Deep and Large Graph Convolutional Networks, Chiang et. al, KDD ‘19

---

> > > ### Author Response · Authors · 2021-11-17
> > > **Response to Reviewer mKbp (Part 3)**
> > >
> > > Response to Reviewer mKbp (continued..)
> > >
> > > > **The GNNAutoScale method of Fey et al. uses similar ideas and make uses of stale embeddings to scale up the training of GNNs. From their experimental results, GNNAutoScale achieves much better performance compared to GraphSAITNT, Cluster-GCN, etc. So, I think the authors should provides a more detailed discussion on the difference and connection between IGLU and GNNAutoScale. Moreover, empirical comparison between them would make the results more complete and convincing.**
> > >
> > > Please refer to our response to the previous question and Section A.6 of the Appendix in the revised version for a detailed comparison of IGLU with VR-GCN, MVS-GNN and GNNAutoScale in terms of methodology and technical novelty.
> > >
> > > ### **Empirical Comparison with GNNAutoScale**
> > >
> > > We have now included an empirical comparison with GNNAutoScale in Section A.6.5 of the Appendix. (Page 20-21). We summarize the results in the table below, for convenience, while the convergence performance plots are included in Figure 6, page 21 of the Appendix.
> > >
> > > It is important to note that the best results for GNNAutoScale as reported by the authors in the paper, correspond to varying hyperparameters such as number of GNN layers and different embedding dimensions across methods, datasets and architectures. However, for the experiments covered in the main paper, we use 2 layer settings for PPI-Large, Flickr and Reddit and 3 layer settings for OGBN-Arxiv and OGBN-Proteins datasets consistently for IGLU and the baseline methods, as motivated by literature. We also ensure that the embedding dimensions are uniform across IGLU and the baselines. Therefore, to ensure a fair comparison, we perform additional experiments with these parameters for GNNAutoScale set to values that are consistent with our experiments for IGLU and the baselines. We train GNNAutoScale with three variants, namely GCN, GCNII and PNA and report the results for each of the variants. We also note here that GNNAutoScale was implemented in PyTorch while IGLU was implemented in TensorFlow. While this makes a wall-clock time comparison unsuitable as discussed in Appendix Section B.2, we provide a wall-clock time comparison for completeness. We also include the best performance numbers for GNNAutoScale on these datasets (as reported by the authors in Table 5, Page 9 of the GNNAutoScale paper) across different architectures.  Note that we do not provide comparisons on the OGBN-Proteins dataset since we ran into errors while trying to incorporate the dataset into the official implementation of GNNAutoScale.
> > >
> > >
> > > **Table:** Test Accuracy of IGLU compared to GNNAutoScale. * - denotes the experiments performed by us using GNNAutoScale in a setting identical to IGLU with 2-layer models on PPI-Large, Reddit and Flickr datasets and 3-layer models on the OGBN-Arxiv dataset (transductive) . For completeness, we also include the best results from GNNAutoScale (GAS-GCN, GCNII, PNA) taken from Table 5, Page 9 of the paper, for comparison. We were unable to perform experiments with GNNAutoScale on the OGBN-Proteins dataset, and hence omit it for comparison.
> > >
> > > | **Algorithm**    | **PPI-Large** | **Reddit** | **Flickr** | **OGBN-Arxiv(trans)** |
> > > | ----------- | ----------- |----------- |----------- |----------- |
> > > | GAS - GCN* | 0.983 | 0.954 | 0.533 | 0.710 |
> > > | GAS - GCNII* | 0.969 | 0.964 | 0.539 | 0.724 |
> > > | GAS - PNA* | 0.917 | 0.970 | 0.555 | 0.714 |
> > > | GAS - GCN | 0.989 | 0.954 | 0.540 | 0.716 |
> > > | GAS - GCNII | 0.995 | 0.967 | 0.562| 0.730 |
> > > | GAS - PNA | 0.994 | 0.971 | 0.566 | 0.725 |
> > > | **IGLU** | 0.987 $\pm$ 0.004 | 0.964 $\pm$ 0.001 | 0.515 $\pm$ 0.001 | 0.719 $\pm$ 0.002 |
> > >
> > >
> > >
> > > ### **Summary of Results:**
> > >
> > > Figure 6 (page 21 of the paper) provides convergence plots comparing IGLU with the different architectures of GNNAutoScale and the Table above summarizes the test performance on PPI-Large, Flickr, Reddit and OGBN-Arxiv (transductive) datasets. From the table we observe that IGLU offers competitive performance compared to the GCN variant of GAS for the majority of the datasets. We also observe from Figure 6 that IGLU offers significant improvements in training time with rapid early convergence on the validation set. Using Equation (1), (Page 7 in the main paper) we quantify these speedups that IGLU offers to reach high validation scores over GAS-GCN and observe that IGLU offers a speedup of 59.45% in reaching a validation score of 0.96 on PPI, a speedup of 97.91% in reaching a validation score of 0.95 on Reddit, a speedup of 88.65% on reaching a validation score of 0.51 on Flickr and a speedup of 39.61% on reaching a validation score of 0.71 on Arxiv (trans) datasets.
> > >
> > > We note that more complex architectures such as GCNII and PNA offer improvements in performance to GNNAutoScale. IGLU being architecture agnostic can be incorporated with these architectures for further improvements in performance. We leave this as an avenue for future work.

---

> > > > ### Comment · Reviewer_mKbp · 2021-11-30
> > > > **Updated score**
> > > >
> > > > Dear authors,
> > > > I really appreciate the detailed response to my comments. The additional experiments on GNNAutoScale makes the work more complete and convincing. As such, I increase my score to 6.

---

> > > > > ### Author Response · Authors · 2021-11-30
> > > > > **Response to Reviewer mKbp**
> > > > >
> > > > > Dear Reviewer mKbp,
> > > > >
> > > > > We would like to thank you for the constructive feedback and for reconsidering the scores. If there are any other questions/feedback for us, we will be very happy to address them before the discussion window closes.

---

> ### Author Response · Authors · 2021-11-19
> **Follow-up on response to Reviewer mKbp**
>
> Dear Reviewer mKbp,
>
> We would like to thank you for your insightful comments and suggestions that have significantly improved the paper.
> We hope that our responses and revision address your concerns satisfactorily and will lead to a more positive evaluation of our work. With the discussion window closing soon, we wanted to know if there are any further clarifications that we can provide from our end.

---

### Official Review · Reviewer_tDRJ · 2021-11-02

**Correctness:** 4
**Technical Novelty And Significance:** 4
**Empirical Novelty And Significance:** 3
**Recommendation:** 6
**Confidence:** 3

**Main Review:**

# Strengths
* I like the idea of holding some variables constant in gradients and using stale computations to speed up computations. Somehow similar ideas have previously been used to, e.g., speed up the convergence of SGD (1) and SGLD (2).
* Strong empirical results show that IGLU scales well and achieves SOTA.
* The authors provide a convergence analysis of IGLU.

# Weaknesses
* In practice, many tricks are used in conjunction with GNNs. Some examples are long-range residual connections, BatchNorm, and virtual nodes. These are especially important to train deeper networks. It is not clear how IGLU could incorporate such tricks. Can the authors elaborate?


1: https://proceedings.neurips.cc/paper/2013/file/ac1dd209cbcc5e5d1c6e28598e8cbbe8-Paper.pdf

2: https://proceedings.neurips.cc/paper/2016/file/9b698eb3105bd82528f23d0c92dedfc0-Paper.pdf





**Summary Of The Paper:**

The work proposes IGLU, an algorithm to scale-up GNNs using stale computations instead of traditional neighborhood sampling.
IGLU has bounded bias if the loss and activation functions are smooth. Results on large-scale benchmarks show IGLU achieves SOTA and scales better than previous methods.

**Summary Of The Review:**

I appreciate the use of stale gradients to speed-up backward computations in GNNs. In principle, the IGLU is architecture-agnostic and can be applied to many GNNs. Therefore, I am more inclined to acceptance than rejection.

---

> ### Author Response · Authors · 2021-11-17
> **Response to Reviewer tDRJ**
>
> > **Concern regarding using Residual Connections, Batch Normalization and Virtual Nodes for training deeper networks**
>
> We thank the reviewer for the comment. Indeed, general purpose techniques such as BatchNorm and skip/residual connections, and GCN-specific advancements such as bi-level aggregation using virtual nodes do offer performance boosts. The current implementation of IGLU already incorporates BatchNorm as described in Section 3. We have included additional content to the paper in Section A.5 (Page 16-18) of the Appendix to demonstrate with examples how all these aforementioned architectural variations can be incorporated into IGLU with minimal changes to the definitions and the guarantees.

---

> ### Author Response · Authors · 2021-11-19
> **Follow up on response to Reviewer tDRJ**
>
> Dear Reviewer tDRJ,
>
> We would like to thank you for your insightful comments and suggestions that have significantly improved the paper.
> We hope that our responses and revision address your concerns satisfactorily and will lead to a more positive evaluation of our work. With the discussion window closing soon, we wanted to know if there are any further clarifications that we can provide from our end.

---

> > ### Author Response · Authors · 2021-11-30
> > **Gentle follow-up to Reviewer tDRJ**
> >
> > Dear Reviewer tDRJ,
> >
> > We would like to thank you for the constructive feedback on our paper. We have tried to incorporate the feedback in the latest draft, hopefully this positively impacts your opinion of the paper.
> >
> > Gentle follow up on the responses and if there are any further questions/concerns we can address with the discussion window closing in a couple of hours.

---

### Author Response · Authors · 2021-11-17
**Summary of rebuttal and revision**

Dear Reviewers, we would like to thank you for your reviews and insightful comments.
We summarize the revisions to the paper during the rebuttal process below: (indicated in blue in the revised version of the paper, for clarity.)

1. We discuss in detail the incorporation of architectural modifications such as Residual Connections, Batch Normalization and Virtual Nodes into IGLU in Section A.5 of the Appendix. (Page 16-18)
2. We compare and contrast IGLU with caching based related works, namely VR-GCN, MVS-GNN and GNNAutoScale in Section A.6.1- A.6.4 of the Appendix and provide an empirical comparison with GNNAutoScale in Section A.6.5 of the Appendix. (Page 18-21)
3. We provide a detailed description of IGLU’s Algorithm 1 and Algorithm 2 in Section A.7 of the Appendix. (Page 21-22)
4. We provide a finer analysis of IGLU’s performance on the OGBN-Proteins dataset in Section A.8 of the Appendix. (Page 22-23)

---

### Decision · Program_Chairs · 2022-01-20

**Decision:**

Accept (Poster)

**Comment:**

Overall the paper present the idea of caching and using stale information to update instead of sub sampling for speeding up graph convolution neural network. Reviewers liked the idea but also there were concerns about experimental comparisons.  In the rebuttal the authors did provide more evidence of comparison with other caching based and other relevant baselines. Overall the importance of scaling up GCNN and empirical results helped the paper cross the high bar.